# Structural insights into the selective recognition of RF-amide peptides by neuropeptide FF receptor 2

Jeesoo Kim [1,2], Sooyoung Hong [1], Hajin Lee[3], Hyun Sik Lee[1], Chaehee Park [1,4], Jinuk Kim[1,5], Wonpil Im [6] & Hee-Jung Choi [1]✉

## Abstract

**Neuropeptide FF Receptor 2 (NPFFR2), a G-protein-coupled receptor, plays a role in pain modulation and diet-induced thermogenesis. While NPFFR2 is strongly activated by neuropeptides FF (NPFFs), it shows low activity in response to RF-amide-related peptides (RFRPs), despite the peptides belonging to a shared family. In contrast, NPFFR1, which shares high sequence similarity with NPFFR2, is activated by RFRPs and regulates reproductive hormone balance. The molecular basis for these receptor-specific interactions with their RF-amide peptides remains unclear. Here, we present cryo-electron microscopy structures of NPFFR2 in its active state bound to the agonist RF-amide peptide hNPSF, and in its ligand-free state. Structural analysis reveals that the C-terminal RF-amide moiety engages conserved residues in the transmembrane domain, while the N-terminal segment interacts in a receptor subtype-specific manner. Key selectivity-determining residues in NPFFR2 are also identified. A homology model of NPFFR1 bound to RFRP, supported by mutagenesis studies, further validates this selectivity mechanism. Additionally, structural comparison between the inactive and active states of NPFFR2 suggests a TM3-mediated activation mechanism. These findings provide insights into RF-amide peptide recognition by NPFF receptors.**

**Keywords** RF-amide Peptide; Neuropeptide FF Receptors; Ligand Selectivity; cryo-EM Structure
**Subject Categories** Membranes & Trafficking; Neuroscience; Structural Biology

## Introduction

The RF-amide neuropeptide family, characterized by the conserved C-terminal Arg-Phe-$NH_2$ (RF-amide) motif, includes a diverse group of peptides such as neuropeptide FFs (NPFFs: NPFF, NPSF, NPAF), RF-amide-related peptides (RFRPs: NPVF/RFRP-1, GnIH/ RFRP-3), prolactin-releasing peptides (PrRPs), pyroglutamylated RF-amide peptides (QRFPs), and kisspeptins (Quillet et al, 2016). These neuropeptides exert their physiological effects by binding to their respective G-protein-coupled receptors (GPCRs): NPFF receptor 2 (NPFFR2, also known as GPR74), NPFF receptor 1 (NPFFR1, also known as GPR147), PrRP receptor (PRLHR), QRFP receptor (QRFPR), and KISS1 receptor (KISS1R), respectively.

Notably, among these five RF-amide peptides, NPFFs and RFRPs share the C-terminal four amino acids, resulting in cross-reactivity between NPFFR2 and NPFFR1, albeit with differing binding affinities. In contrast, other RF-amide peptides have variable sequences and lengths except for the conserved RF-amide motif, resulting in specific interactions with their respective receptors. Recent cryo-EM structures of QRFP26–QRFPR–$G_{sqi}$ (PDB: 8ZH8) (Iwama et al, 2024), Kisspeptin-10–KISS1R–$G_{qNi}$ (PDB: 8ZJD) (Shen et al, 2024), and PrRP20–PrRPR–$G_{sqi}$ (PDB: 8ZPT) (Li et al, 2024) have revealed that while the C-terminal RF-amide motif is integral to their binding modes, QRFP26, Kisspeptin-10, and PrRP20 display distinct receptor-binding interactions beyond this motif. These findings underscore the diverse molecular mechanisms underlying RF-amide peptide–receptor interactions.

Among the five RF-amide receptors, KISS1R shows notable phylogenetic divergence, with relatively low sequence similarity to the other four RF-amide receptors (~33%) (Appendix Fig. S1). Interestingly, neuropeptide Y (NPY) receptors, although not part of the RF-amide receptor family, are more closely related to the four RF-amide receptors than KISS1R is, sharing ~50% sequence similarity. By contrast, NPFFR1 and NPFFR2 are the most closely related, exhibiting 68% sequence similarity, yet they display distinct peptide binding and activation profiles toward NPFFs and RFRPs. NPFFR1 preferentially binds NPVF/RFRP-1 and GnIH/RFRP-3, while NPFFR2 favors NPSF, NPFF, and NPAF. These differences in peptide selectivity reflect their distinct physiological functions: NPFFR2 is involved in modulating pain, cardiovascular regulation, and diet-induced thermogenesis (Kotlinska et al, 2007; Wu et al, 2010; Zhang et al, 2022), whereas NPFFR1 primarily regulates gonadotropin release and reproductive hormone balance (Lents et al, 2016; León et al, 2014; Ubuka and Tsutsui, 2014).

Since NPFF has been shown to be involved in opioid tolerance and opioid-induced hyperalgesia (Oberling et al, 1993), drug

---

[1]Department of Biological Sciences, Seoul National University, Seoul 08826, Republic of Korea. [2]Institute for Data Innovation in Science, Seoul National University, Seoul 08826, Republic of Korea. [3]MolCube, Inc., Seoul 06640, Republic of Korea. [4]The Research Institute of Basic Sciences, Seoul National University, Seoul 08826, Republic of Korea. [5]Division of Biological Science and Technology, Yonsei University, Wonju 26493, Republic of Korea. [6]Department of Biological Sciences, Lehigh University, Bethlehem, PA 18015, USA. ✉E-mail: choihj@snu.ac.kr

development targeting NPFFR2 holds significant potential for treating opioid-related pathological conditions, such as antinociception, tolerance, dependence, and hyperalgesia. For instance, RF9, which is classified as a partial agonist, often exhibits an antagonist-like effect in some experimental models, demonstrating that blockade of NPFFR2 in rat brain is effective in treating opioid-induced hyperalgesia and withdrawal symptoms, while also mitigating side effects like opioid-induced constipation (Simonin et al, 2006). Conversely, NPFFR2 agonists such as 1DMe have shown potent antinociceptive effects, reducing morphine tolerance and reversing inflammatory hyperalgesia in rat following intrathecal injection (Courteix et al, 1999). Recently, multitarget drugs that act on both opioid receptors and NPFFR2 have shown promising results in preclinical studies (De Neve et al, 2024; Li et al, 2016). In drug development targeting NPFFR2, careful consideration of off-target effects is crucial. Since NPFFR1 and NPFFR2 exhibit cross-reactivity toward each other's ligands, even though it is limited, it is essential to understand the structural factors that determine each receptor's specificity to design more selective drugs.

Despite the biological and therapeutic significance of specific recognition of NPFFs by NPFFR2, the underlying molecular basis for these specific interactions has remained elusive. Here, we present the structure of human NPSF-bound NPFFR2 in complex with $G_i$ using single-particle cryo-electron microscopy (cryo-EM). Structural analysis of the ligand-binding site of NPFFR2 provides molecular insights into the selective recognition of RF-amides by NPFFR2. To further investigate the selectivity of NPFFR1, we generated a homology model based on our NPFFR2 structure and validated the model through mutagenesis studies. In addition, we present a ligand-free structure of NPFFR2, revealing its inactive state, which enables us to understand the activation mechanism of NPFFR2 at the molecular level. Altogether, our study sheds light on the mechanisms underlying the specific recognition of RF-amides by NPFFRs and their subsequent activation.

# Results

## Overall structure of agonist-bound NPFFR2 coupled with $G_i$

Several isoforms of NPFF peptides are known: NPAF (AGEGLNSQFWSLAAPQRF-NH$_2$), NPSF (SQAFLFQPQRF-NH$_2$, also called SQA-NPFF) (Perry et al, 1997; Sol et al, 1999), and NPFF (FLFQPQRF-NH$_2$) (Appendix Table S1). These peptides are well-known agonists for NPFFR2 (Elshourbagy et al, 2000). Among them, the eight-residue core sequence of NPFF is highly conserved across mammals, while the N-terminal extension of NPSF varies among species. To ensure clarity, we will refer to the human form of NPSF (SQA-NPFF) as hNPSF throughout this study.

Using forskolin-stimulated cAMP assays, we confirmed that hNPSF exhibits approximately 5-fold higher potency than the octapeptide NPFF (Fig. 1A; Table EV1). This result, consistent with previously published data (Elshourbagy et al, 2000), suggests that hNPSF represents the minimum length of peptide that achieves maximum potency for NPFFR2 among these peptides. Thus, we selected hNPSF as the optimal agonist for our structural study of the agonist-bound NPFFR2 in its active state. To obtain the complex of hNPSF, human NPFFR2, and $G_i$, we separately purified NPFFR2 and

heterotrimeric $G_i$ and then assembled them in the presence of hNPSF and scFv16, which stabilizes the $G_i$ complex. The structure of this complex was determined by cryo-EM at a global resolution of 3.2 Å (Figs. 1B and EV1; Appendix Table S2).

The overall structure of hNPSF-bound NPFFR2 adopts the canonical active conformation of a class A GPCR coupled to a Gprotein. Interestingly, the hNPSF–NPFFR2–$G_i$ complex exhibits high structural similarity to the NPY–NPY2R–$G_i$ complex (Kang et al, 2023), with an RMSD of 0.9 Å for their transmembrane domains (TMDs) (Fig. EV2A). However, distinct structural differences are observed in the ECL2 regions and N-terminal tails of the two receptors. In the NPY-bound structure of NPY2R, the ECL2 forms a β-hairpin that engages in hydrophobic interactions with the helical region of NPY. By contrast, the ECL2 of NPFFR2 consists of four β-strands, β2-β5, and an additional β1 strand is formed by residues 32-36 in the N-terminal tail. Notably, β3 bends approximately 90 degrees from β2 and forms a β-sheet with β1 and β4, while β2 and β5 form another distinct β-sheet (Fig. 1C). In both structures, the ECL2 is stabilized by a conserved disulfide bond between Cys$^{3.25}$ on TM3 and Cys$^{45.50}$ on ECL2 (Fig. 1C).

NPFFR2 preferentially couples with $G_{i/o}$ (Mollereau et al, 2002), while QRFPR and KISS1R are known to favor $G_q$ coupling (Kotani et al, 2001). The binding interface between NPFFR2 and $G\alpha_i$ measures 774 Å$^2$, in contrast to the more extensive 1455 Å$^2$ interface observed in the KISS1R and $G\alpha_{qNi}$ complex. In KISS1R, the cytoplasmic regions of TMD and intracellular loop 2 (ICL2) are involved in the interactions with the αN and α5 regions, and the α4/β6 regions of the $G\alpha_q$ subunit. By comparison, the interaction between NPFFR2 and $G\alpha_i$ is primarily mediated through the α5 helix of the $G\alpha_i$ subunit and the TMD of the receptor. The last five amino acids of the α5 helix exhibit sequence variations across different Gα subunits, resulting in distinct interaction patterns with GPCRs. In the structure of NPFFR2 coupled to $G\alpha_i$, F354$^{H5.26}$ of $G\alpha_i$ interacts with NPFFR2 via van der Waals contacts with the Cβ carbon of K268$^{6.29}$; however, its side chain is not fully resolved in our structure. In contrast, the corresponding residue V361$^{H5.26}$ in $G\alpha_{qNi}$ does not establish any contact with KISS1R. Instead, N359$^{H5.24}$ and Y358$^{H5.23}$ of $G\alpha_{qNi}$ engage in extensive interactions with KISS1R; the former makes polar contacts with the backbone carbonyl of Y323$^{7.53}$ and the amide group of F330$^{8.50}$, and the latter interacts with T77$^{2.39}$, Y80$^{2.43}$, M136$^{3.46}$, D139$^{3.49}$, and R140$^{3.50}$. In contrast, the corresponding residues in $G\alpha_i$, G352$^{H5.24}$, and C351$^{H5.23}$, exhibit minimal interactions with NPFFR2; the former lacks any contact with the receptor, and the latter forms only a van der Waals interaction with C146$^{3.53}$ of NPFFR2 (Appendix Fig. S2).

## Recognition of RF-amide peptide by NPFFR2

Interactions of hNPSF with the NPFFR2 are described by dividing the peptide into three parts, core interaction region (PQRF, positions −1 to −4, numbered from −1 at the C-terminus) (Wu et al, 2024), specificity determining region (FLFQ, positions −5 to −8), and the extra N-terminal region (SQA, positions −9 to −11) (Fig. 2A).

As the C-terminus of hNPSF enters to the TMD binding pocket, F(−1)$^{SF}$ (superscript SF refers to the residue from hNPSF) is located ~12 Å deep from the membrane surface. While the C-terminal amide group forms polar contacts with T102$^{2.61}$, Q125$^{3.32}$, and H319$^{7.39}$, the phenyl group is positioned into the hydrophobic pocket formed by V129$^{3.36}$, L223$^{5.42}$, and W291$^{6.52}$ (Fig. 2A; Appendix Fig. S3A). The

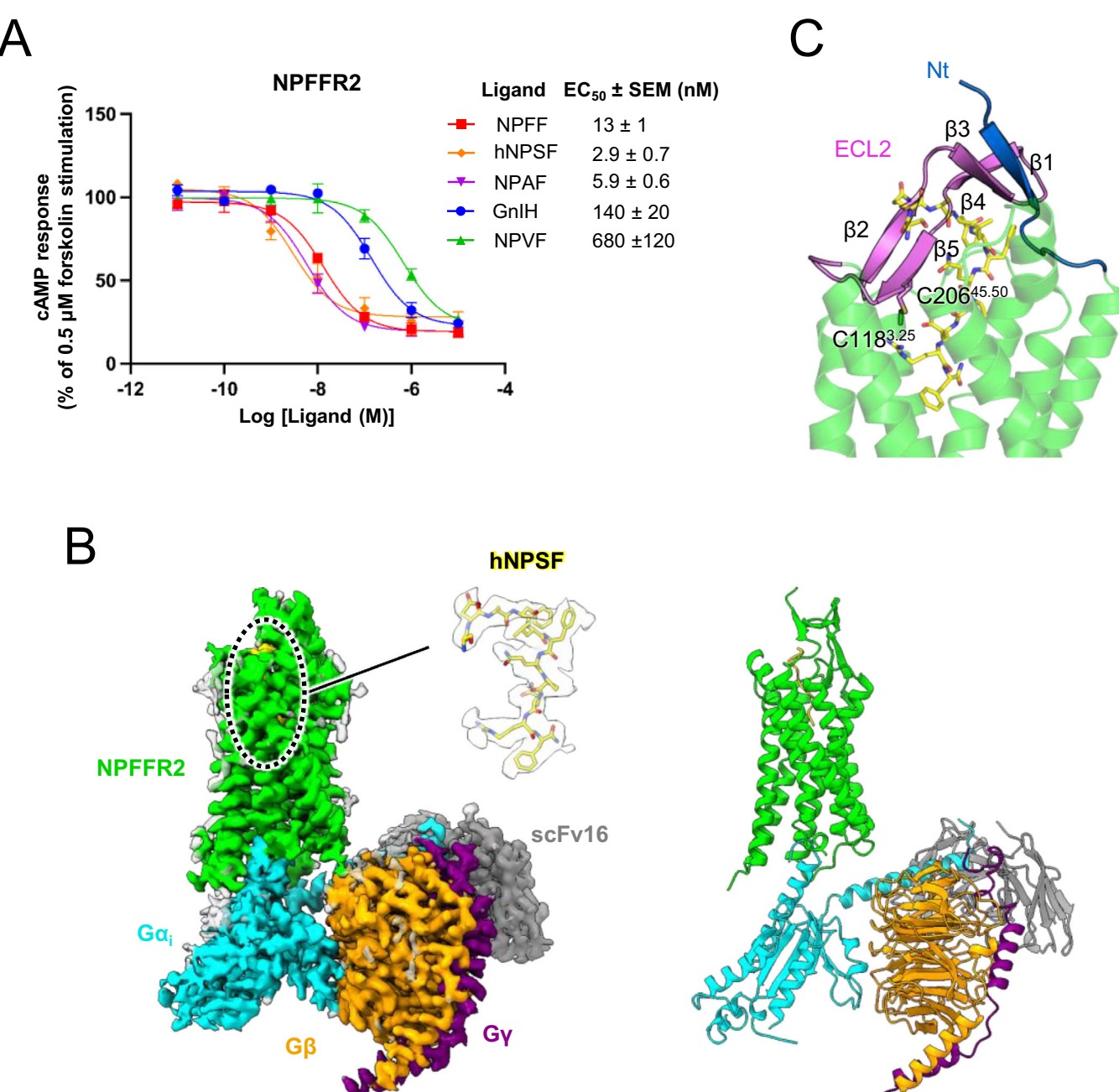

**Figure 1. Structural analysis and downstream signaling of NPFFR2.**

(A) The concentration-dependent response of various agonists on NPFFR2 is measured using a forskolin-stimulated cAMP assay. Data points represent mean values from three independent experiments (each with technical triplicate), with error bars indicating SEM (standard error of the mean). Raw data were normalized to the vehicle-treated (0%) and 0.5 μM forskolin-treated (100%) responses. Emax and EC$_{50}$ values along with their respective errors are summarized in Table EV1. (B) The overall cryo-EM electron density map and structural model of the hNPSF–NPFFR2–Gα$_i$βγ–scFv16 complex are presented, with the following color scheme: hNPSF (yellow), NPFFR2 (green), Gα$_i$ (cyan), Gβ (orange), Gγ (purple), and scFv16 (gray). (C) The extracellular loop 2 (ECL2, pink) and N-terminus (Nt, blue) of NPFFR2 are depicted using a cartoon model. They form stable β-sheets. Source data are available online for this figure.

importance of these residues for downstream signaling is demonstrated by the significant loss of G$_i$ signaling of the relevant Ala mutants (Fig. 2B; Table EV1; Appendix Figs. S4 and S5). It is interesting to note that the C-terminal Tyr of NPY is positioned very similarly to the C-terminal Phe of RF-amide in the receptor, having similar polar interactions of the C-terminal amide of NPY with

T107$^{2.61}$, Q130$^{3.32}$, and H311$^{7.39}$ of NPY2R (Fig. EV2B). In contrast, residues constituting binding pocket for Y(−1)$^{NPY}$ of NPY in NPY2R are less hydrophobic. In particular, A130$^{3.37}$, L223$^{5.42}$, and W291$^{6.52}$ of NPFFR2 are replaced with Q135$^{3.37}$, S223$^{5.42}$, and H285$^{6.52}$ in NPY2R, which forms polar contacts with the hydroxyl group of Tyr of NPY (Fig. EV2B). The difference in hydrophobicity of the C-terminal

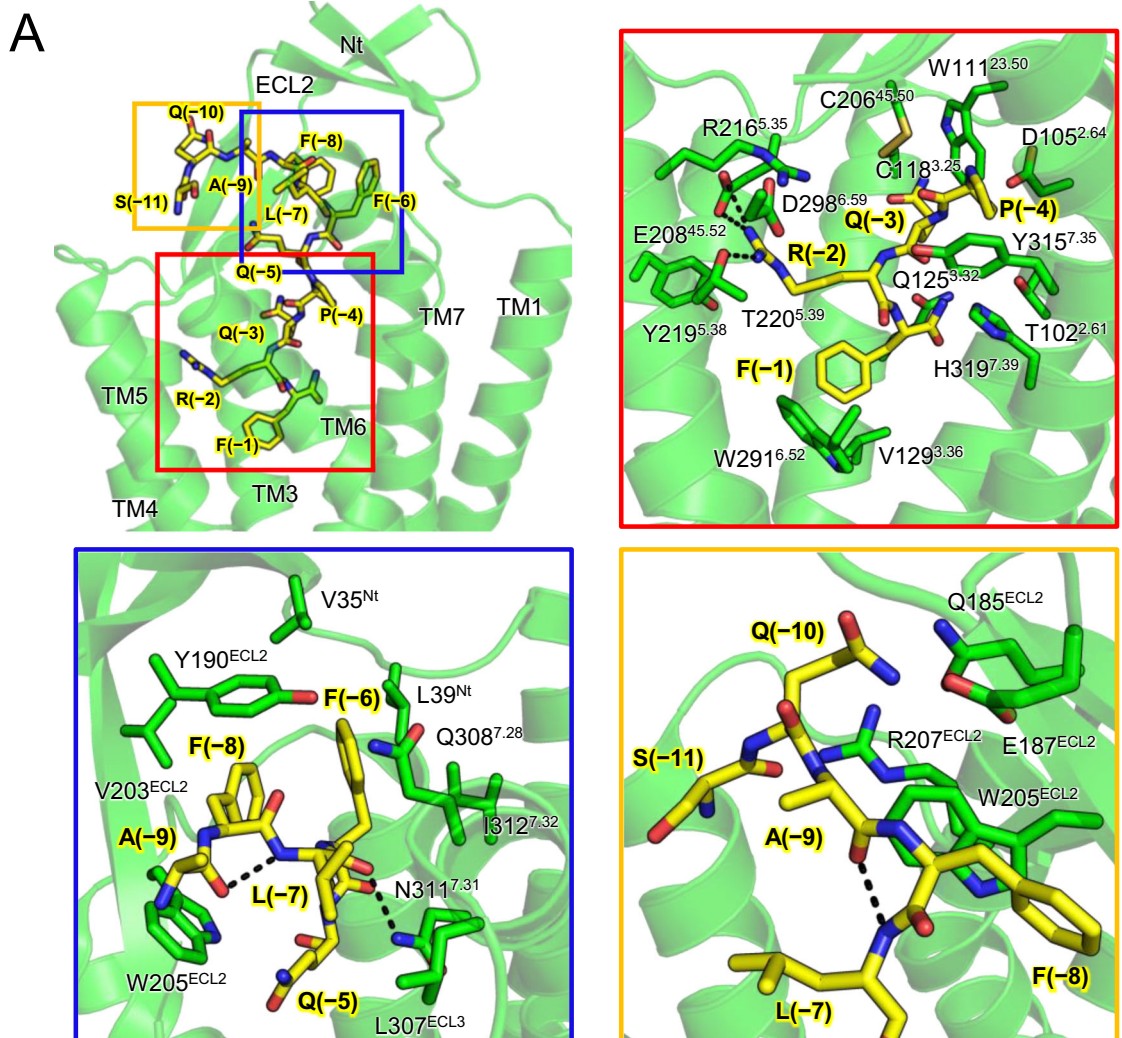

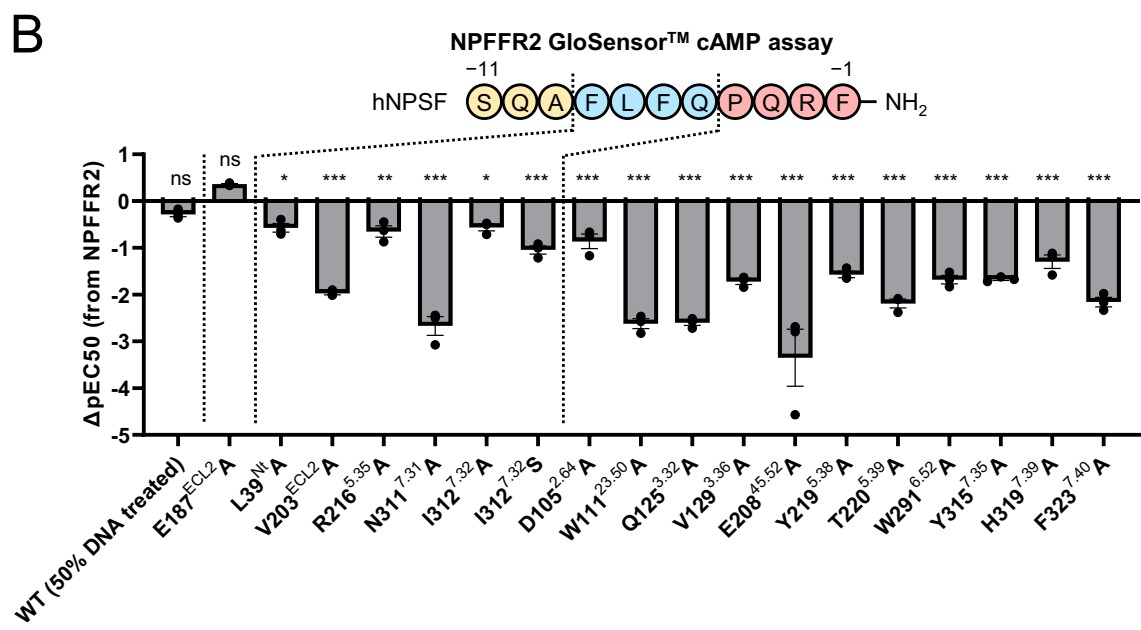

**Figure 2. Interaction of hNPSF with NPFFR2.**

(A) Residues participating in the interactions between hNPSF and NPFFR2 in the transmembrane (TM) binding pocket are shown as sticks. Dashed lines indicate polar interactions and hydrogen bonds between residues. Zoomed-in views highlighting interactions of the C-terminal segment (residues −1 to −4, PQRF), middle segment (residues −5 to −8, FLFQ), and the N-terminal segment (residues −9 to −11, SQA) of hNPSF are enclosed in red, blue, and yellow boxes, respectively. (B) NPFFR2 residues interacting with hNPSF were mutated to alanine, and the impact of each mutation on the $EC_{50}$ value ($\Delta pEC_{50}$) was investigated using a forskolin-stimulated cAMP assay, after confirming surface expression of each mutant by an ELISA-based surface expression assay (Appendix Fig. S5). Three independent experiments (with technical triplicates) were conducted for each mutant. The bar graphs represent the mean of $\Delta pEC_{50}$ values from each experiment, with error bars indicating SEM. Emax and $EC_{50}$ values along with their respective errors are summarized in Table EV1. Statistical analyses were performed using ordinary one-way ANOVA followed by Dunnett's test, comparing to the wild-type response. ns (not significant, $P > 0.05$); *$P < 0.05$; **$P < 0.01$; ***$P < 0.001$. The exact $P$ values are provided in Table EV1. Source data are available online for this figure.

residues of RF-amide and NPY, Phe vs. Tyr, appears to contribute to lack of cross-activity between NPFFR2 and NPY2R, as NPY and hNPSF exhibit almost no activity towards NPFFR2 and NPY2R, respectively (Appendix Fig. S6). $R(-2)^{SF}$ of RF-amide binds to the receptor through a salt bridge with $E208^{45.52}$ and a hydrogen bond with $T220^{5.39}$ (Fig. 2A). Mutations of $E208^{45.52}$ and $T220^{5.39}$ into Ala resulted in 1,000-fold and 200-fold reduced downstream signaling, respectively (Fig. 2B and Table EV1), suggesting that these interactions are important for RF-amide ligand binding. In addition, $Y219^{5.38}$ stabilizes $R(-2)^{SF}$ through a cation-π interaction. $R(-2)^{SF}$ of RF-amide is also positioned similarly to $R(-2)^{NPY}$ of NPY, which forms electrostatic interactions and polar contacts with $E205^{45.52}$, $S220^{5.39}$, and $D292^{6.59}$ of NPY2R. Compared to RF-amide region, $Q(-3)^{SF}$ and $P(-4)^{SF}$ exhibit less extensive contacts with the receptor, the former interacts with $W111^{23.50}$ and the disulfide-forming $C206^{45.50}$, the latter forms van der Waals contacts with $D105^{2.64}$ and $Y315^{7.35}$ (Fig. 2A).

The middle segment of the peptide, FLFQ, corresponds to the N-terminal 4 amino acids of NPFF. The two Phe residues, $F(-8)^{SF}$ and $F(-6)^{SF}$ stack together, forming a hydrophobic interaction network with $V35^{Nt}$, $L39^{Nt}$, $Y190^{ECL2}$, $V203^{ECL2}$, $W205^{ECL2}$, and $I312^{7.32}$ of the receptor (Fig. 2A; Appendix Fig. S3B). Mutations of these residues to Ala resulted in reduced signaling, demonstrating the importance of these residues for hNPSF binding and downstream signaling (Fig. 2B and Table EV1). Another hydrophobic residue $L(-7)^{SF}$ interacts with $L307^{ECL3}$ and the methylene group of $Q308^{7.28}$. In contrast, the polar residue $Q(-5)^{SF}$ is surrounded by polar residues of the receptor, $R216^{5.35}$, $S297^{6.58}$, and $N311^{7.31}$, but they are 4 to 5 Å away from $Q(-5)^{SF}$, suggesting a lack of direct polar interactions. However, as these residues are readily accessible from the extracellular milieu, water molecules are likely involved in water-mediated polar interactions. Because of the resolution limitation of our structure, we could not observe the captured water molecules in this region, if they are. Instead, MD simulations demonstrated that water molecules are observed between $Q(-5)^{SF}$ and polar residues of NPFFR2, such as $R216^{5.35}$, $S297^{6.58}$, $N311^{7.31}$, and $Y315^{7.35}$, mediating polar interactions among them. (Fig. EV3; Appendix Fig. S7).

The N-terminal three residues, Ser-Gln-Ala, which are missing in NPFF do not form an extensive interaction network with the receptor, explaining that hNPSF display only 5-fold higher potency than NPFF (Fig. 1A; Table EV1). $Q(-10)^{SF}$ is located near $Q185^{ECL2}$, $E187^{ECL2}$, $W205^{ECL2}$, and $R207^{45.51}$ of ECL2, forming polar contacts with them (Fig. 2A; Appendix Fig. S3C). However, single point mutation of each residue into Ala, such as $E187^{ECL2}A$, did not show a dramatic decrease in signaling, suggesting that interaction of each individual residue is weak. (Fig. 2B; Table EV1). Consistently, our MD simulations of the hNPSF–NPFFR2 complex demonstrate that fluctuation of hNPSF

increases from the C-terminus to the N-terminus during a 1.5 μs simulation time, suggesting that the N-terminal SQA does not have stable interactions with the receptor (Appendix Fig. S8). Nevertheless, these N-terminal interactions, though weak and transient, may contribute to the slightly higher signaling activity of hNPSF compared to NPFF (Fig. 1A; Table EV1).

## Structural comparison with other RF-amide receptors

Recently, the cryo-EM structures of the QRFP26–QRFPR–$G_{sqi}$ (PDB: 8ZH8), Kisspeptin-10–KISS1R–$G_{qNi}$ (PDB: 8ZJD), and PrRP20–PrRPR–$G_{sqi}$ (PDB: 8ZPT) complexes have been reported (Iwama et al, 2024; Li et al, 2024; Shen et al, 2024). Structural alignment of the TMDs of NPFFR2 with those of these RF-amide receptors revealed RMSD values of 1.1 Å for QRFPR, 1.3 Å for KISS1R, and 1.0 Å for PrRPR, indicating high structural similarity in their TMDs. In contrast, these receptors display distinct structural features in their ECL2s and N-terminal regions (Fig. 3A). In QRFPR, the ECL2 forms a β-sheet structure, and its N-terminus exhibits a unique helix-loop-helix (HLH) configuration that extends outside the extracellular region. This structure is crucial for the interaction with the helical domain of its ligand, QRFP26. Conversely, the ECL2 of KISS1R and PrRPR are relatively short, and their N-termini are flexible.

The five subtypes of RF-amide receptors commonly recognize the C-terminal RF-amide of their peptide ligands, leading to the expectation that the residues interacting with this region are conserved across this family (Fig. 3C). Indeed, the key residues interacting with the amide group—$T^{2.61}$, $Q^{3.32}$, and $H^{7.39}$—as well as those interacting with the side chain of the C-terminal $F(-1)^{RF}$ (superscript RF refers to the residue from RF-amide peptides in general), such as $V^{3.36}$ are mostly conserved according to the Valdar method calculation (Valdar, 2002). The residues interacting with $R(-2)^{RF}$–$E^{45.52}$ and $T^{5.39}$, are also conserved except for KISS1R and PrRPR where $T^{5.39}$ is replaced by alanine. Surprisingly, although $E^{45.52}$ is conserved in KISS1R, it does not interact with $R(-2)^{RF}$ in the Kisspeptin-10–KISS1R–$G_{qNi}$ complex structure. Instead, $R(-2)^{RF}$ of Kisspeptin-10 interacts with $Q123^{3.33}$ (Fig. 3B), whose corresponding residue in NPFFR2 is $G126^{3.33}$. In the QRFP26–QRFPR–$G_{sqi}$ complex structure, the conserved interaction pattern for the C-terminal RF-amide of the peptide is observed (Fig. 3B). Consistently, sequence analysis suggests that NPFFR2 shares more structural similarities with QRFPR than with KISS1R and PrRPR, with TMD sequence identities of 38%, 33%, and 31%, respectively.

Beyond the third position from the C-terminus, the peptide ligand sequences diverge, leading to distinct interaction patterns

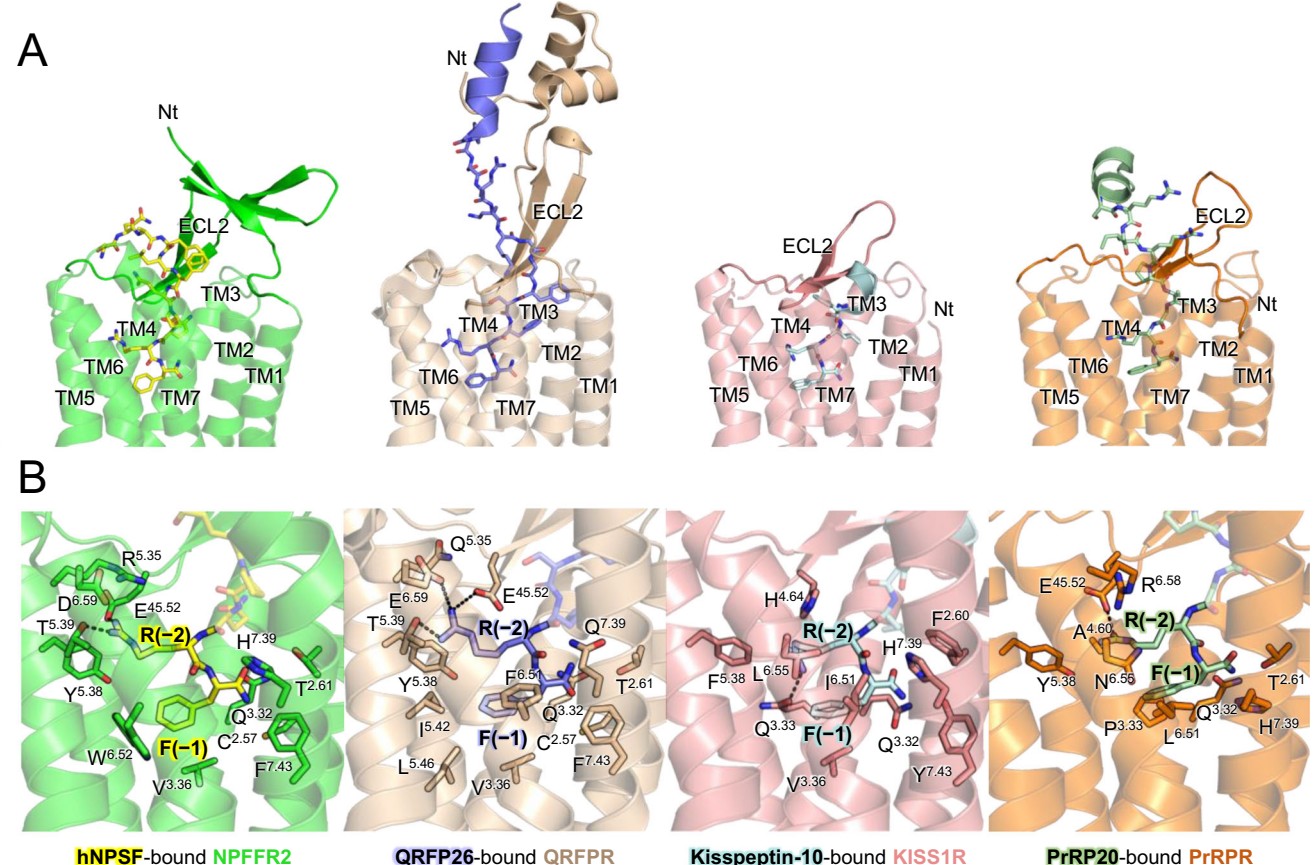

**Figure 3. Structural comparison between RF-amide receptors.**

(A) The endogenous ligand-bound structures of RF-amide receptors, including hNPSF-bound NPFFR2, QRFP26-bound QRFPR (PDB: 8ZH8), Kisspeptin-10-bound KISS1R (PDB: 8ZJD), and PrRP20-bound PrRPR (PDB:8ZPT) are shown. The unique ECL2 structures and ligand-binding modes are depicted with cartoon models: NPFFR2 (green), hNPSF (yellow), QRFP26 (blue-purple), QRFPR (light brown), Kisspeptin-10 (light cyan), KISS1R (salmon), PrRP20 (pale green), and PrRPR (orange). (B) Detailed views of the TM binding pocket of each RF-amide receptor show that these receptors share common binding mode while also exhibiting unique binding interactions with the C-terminal RF-amide motif of their respective ligands. (C) This table illustrates the residues of NPFFR2 and other RF-amide receptors (NPFFR1, QRFPR, KISS1R, and PrRPR) that interact with the RF-amide motif at the C-terminus of their respective ligands. For NPFFR1, the residues shown correspond to those interacting residues in NPFFR2. The shading represents the conservation score of each residue among the RF-amide receptors, calculated using the Valdar method, with darker gray indicating higher conservation. Negatively charged residues are shown in red, positively charged residues in blue, and uncharged residues in white or black.

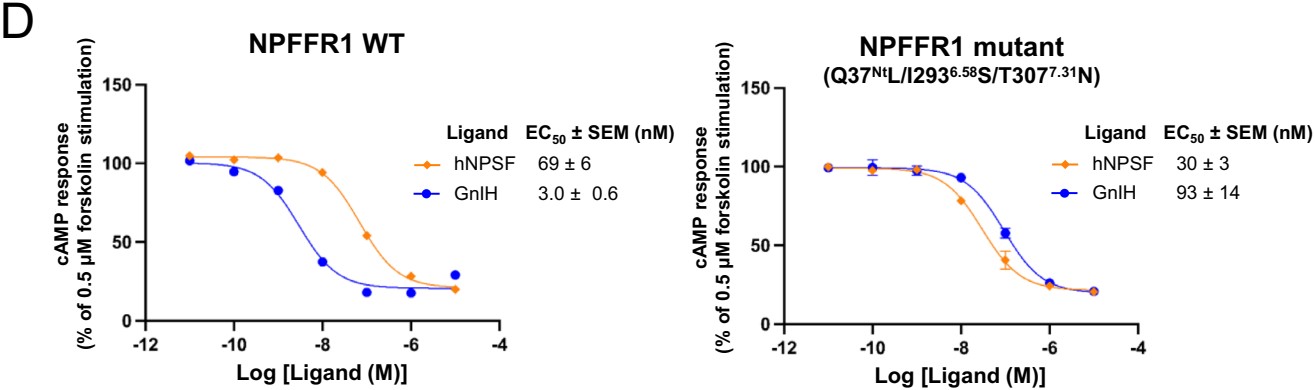

**A**

NPFFs
- NPFF: FL**FQ**PQRF-NH₂
- hNPSF: SQAFL**FQ**PQRF-NH₂
- NPAF: AGEGLNSQFWSLAAPQRF-NH₂

RFRPs
- GnIH: MPHSFA**NL**PLRF-NH₂
- NPVF: VP**NL**PQRF-NH₂

**B**

NPFFR1

| Ligand | EC₅₀ ± SEM (nM) |
|---|---|
| NPFF | 42 ± 5 |
| hNPSF | 69 ± 6 |
| NPAF | 210 ± 40 |
| GnIH | 3.0 ± 0.6 |
| NPVF | 2.0 ± 0.4 |

**C**

hNPSF-bound NPFFR2
GnIH-bound NPFFR1 (AlphaFold2)

hNPSF-bound NPFFR2 | GnIH-bound NPFFR1

hydrophobic — hydrophilic

**D**

NPFFR1 WT

| Ligand | EC₅₀ ± SEM (nM) |
|---|---|
| hNPSF | 69 ± 6 |
| GnIH | 3.0 ± 0.6 |

NPFFR1 mutant
(Q37^Nt L/I293^6.58 S/T307^7.31 N)

| Ligand | EC₅₀ ± SEM (nM) |
|---|---|
| hNPSF | 30 ± 3 |
| GnIH | 93 ± 14 |

**Figure 4. RF-amide ligand selectivity of NPFFR2 and NPFFR1.**

(A) Sequence alignment between human NPFFs and RFRPs, the endogenous ligands of NPFFRs, which share PXRF-NH$_2$ motif at their C-terminus (shown in green letters). Key residues at the fifth and sixth positions from the C-terminus, are highlighted in bold. The alignment clearly shows that RFRPs have hydrophobic residues (black) at the fifth position and hydrophilic residues (red) at the sixth position, while NPFF and hNPSF have reversed hydrophobicity at these key positions. (B) The concentration-dependent response of various agonists on NPFFR1 was measured using a forskolin-stimulated cAMP assay. Data points represent the mean values from three independent experiments (each with technical triplicate), with error bars indicating SEM. Raw data were normalized to the vehicle-treated (0%) and 0.5 μM forskolin-treated (100%) responses. (C) Structural comparison between the binding modes of hNPSF to NPFFR2 and GnIH to NPFFR1. An AlphaFold2 prediction model of the GnIH–NPFFR1 complex was generated using the hNPSF–NPFFR2 complex structure as a template. The hydrophobicity of NPFFRs is shown with surface representation, where purple indicates hydrophilic regions and bronze indicates hydrophobic regions. The upper left panel shows a hydrophobic patch on the NPFFR2 surface where F(−6) of hNPSF binds, while the upper right panel depicts the corresponding hydrophilic surface region of NPFFR1, where N(−6) of GnIH is located. The lower left panel illustrates the hydrophilic surface near the water pocket close to the binding site of Q(−5) of hNPSF, while the lower right panel displays the corresponding hydrophobic region in NPFFR1 where L(−5) of GnIH binds, which is hydrophobic. (D) The cAMP response to the ligands hNPSF and GnIH was measured for both wild-type NPFFR1 and a mutant form of NPFFR1 using a forskolin-stimulated cAMP assay. The NPFFR1 mutant (Q37$^{Nt}$L, I293$^{6.58}$S, and T307$^{7.31}$N) shows an increased EC$_{50}$ for hNPSF while exhibiting a decreased response to GnIH. Symbols represent mean values from three independent experiments, each conducted in technical triplicates, with error bars indicating SEM. Responses were normalized to vehicle-treated controls (0%) and 0.5 μM forskolin-treated controls (100%). Source data are available online for this figure.

with their respective receptors (Appendix Fig. S9). The middle to the N-terminal fragment of hNPSF primarily engages with TM6, TM7, ECL2, and ECL3. QRFP26 interacts with TM2 and TM3 of QRFPR, and its N-terminal helix binds strongly to the HLH of QRFPR's N-terminus. Kisspeptin-10 forms a short helix structure at its N-terminus, interacting with the extracellular regions of TM2 and TM7, as well as ECL1, ECL2, and ECL3. The N-terminal helix of PrRP20 is positioned above the receptor TMD and does not form close contact with PrRPR.

These comparisons indicate that while most RF-amide receptors share a conserved binding mode for the C-terminal RF-amide of peptide ligands, variations in extracellular TM residues and distinct ECL structures drive differences in peptide-binding specificity for the remaining portions of the peptides.

## Key interactions of hNPSF determining the ligand-receptor specificity

RF-amide peptides that act on NPFFR2 are categorized into the NPFF series (NPFF, hNPSF, and NPAF), which strongly activate NPFFR2, and the RFRP series (GnIH, NPVF), which weakly activate NPFFR2 but more effectively activate NPFFR1 (Figs. 1A and 4A,B; Table EV1). The cross-activity of NPFF and RFRP series with NPFFR1 and NPFFR2, though limited, is explained by the high sequence conservation of the C-terminal four amino acids in these peptides, as well as the conserved NPFFR2 residues that interact with the corresponding four amino acids of hNPSF in NPFFR1 (Appendix Fig. S10). Notably, NPFFs and RFRPs, which differ in the residues at position −5 to N-terminus, selectively activate NPFFR2 and NPFFR1, respectively (Figs. 1A and 4A,B). This suggests that these −5 to N-terminal residues are responsible for determining ligand-receptor selectivity. In particular, when sequences of four different RF-amide peptides that bind to NPFFRs are aligned, it is noticeable that the peptides can be divided into two groups depending on the amino acids on the 5th and 6th positions from the C-terminus. That is, while NPFF series has Phe-Gln sequences, RFRPs contains Asn-Leu sequences (Fig. 4A). Since F(−6)$^{SF}$ is located at the hydrophobic pocket formed by V35$^{Nt}$, L39$^{Nt}$, Y190$^{ECL2}$, and I312$^{7.32}$ of NPFFR2, the corresponding residue, Asn of RFRPs would not be preferred in this position (Fig. 4C) In the case of Q(−5)$^{SF}$, it is located within polar environment formed by R216$^{5.35}$, S297$^{6.58}$, N311$^{7.31}$, and Y315$^{7.35}$ of the receptor. Because of this polar environment, Leu of RFRPs would not be fitted well in this position.

To better understand the molecular basis of how NPFFR1 prefers GnIH over NPFF, structural model of the GnIH-bound NPFFR1 complex was generated by AlphaFold2 (Jumper et al, 2021) using our hNPSF-bound NPFFR2 structure as a template. NPFFR1 and NPFFR2 exhibit high sequence similarity in the TMD and ECL2 with overall sequence similarity of 84 and 68%, respectively (Appendix Fig. S10). Consequently, the AlphaFold2 prediction model for NPFFR1 demonstrates high reliability, with a pLDDT score of 79.6. The predicted overall structure of NPFFR1 closely resembles that of NPFFR2 in the ECL2 as well as in the TMD. The predicted binding position of the C-terminal four residues (PLRF) of GnIH is identical to the location of PQRF in the hNPSF–NPFFR2–G$_i$ structure. However, in NPFFR1, the residues corresponding to the water pocket in NPFFR2 (Appendix Fig. S10) are substituted with hydrophobic residues such as I293$^{6.58}$, T307$^{7.31}$ and F311$^{7.35}$ (Fig. 4C). Therefore, NPFFR1 is likely to prefer hydrophobic residues like L(−5)$^{GnIH}$ instead of polar residue like Gln of NPFF (Fig. 4C). In addition, the vicinity of N(−6)$^{GnIH}$ can form polar contacts with NPFFR1, as the residues involved in the hydrophobic patch of NPFFR2 are replaced with polar residues, S33$^{Nt}$, Q37$^{Nt}$, T106$^{ECL1}$ and S202$^{ECL2}$.

To further validate whether these specific residues in NPFFR1 affect ligand selectivity, we generated a triple mutant of NPFFR1, in which residues Q37$^{Nt}$, I293$^{6.58}$, and T307$^{7.31}$ of NPFFR1 were replaced by the corresponding NPFFR2 residues, Leu, Ser, and Asn, respectively (Appendix Fig. S10). Signaling assays were conducted using this mutant in response to hNPSF and GnIH. The results demonstrated a 30-fold decrease in affinity for GnIH, while the affinity for hNPSF increased by approximately 2-fold (Fig. 4D; Table EV1). Consequently, this mutant exhibited reversed selectivity compared to wild-type NPFFR1, with a higher preference for hNPSF over GnIH. This finding supports our hypothesis regarding RF-amide peptide selectivity by the receptor.

## Overall structure of ligand-free state NPFFR2

Structural comparison between inactive and active states of NPFFR2 would provide an activation mechanism of NPFFR2 upon agonist binding. So far, strong antagonist or inverse agonist stabilizing the inactive state of NPFFR2 has not been commercially available, so, we solved the ligand-free state structure of NPFFR2 using cryo-EM analysis. For this study, we replaced the residues 246-266 in the ICL3 region of NPFFR2 with cytochrome b562 RIL

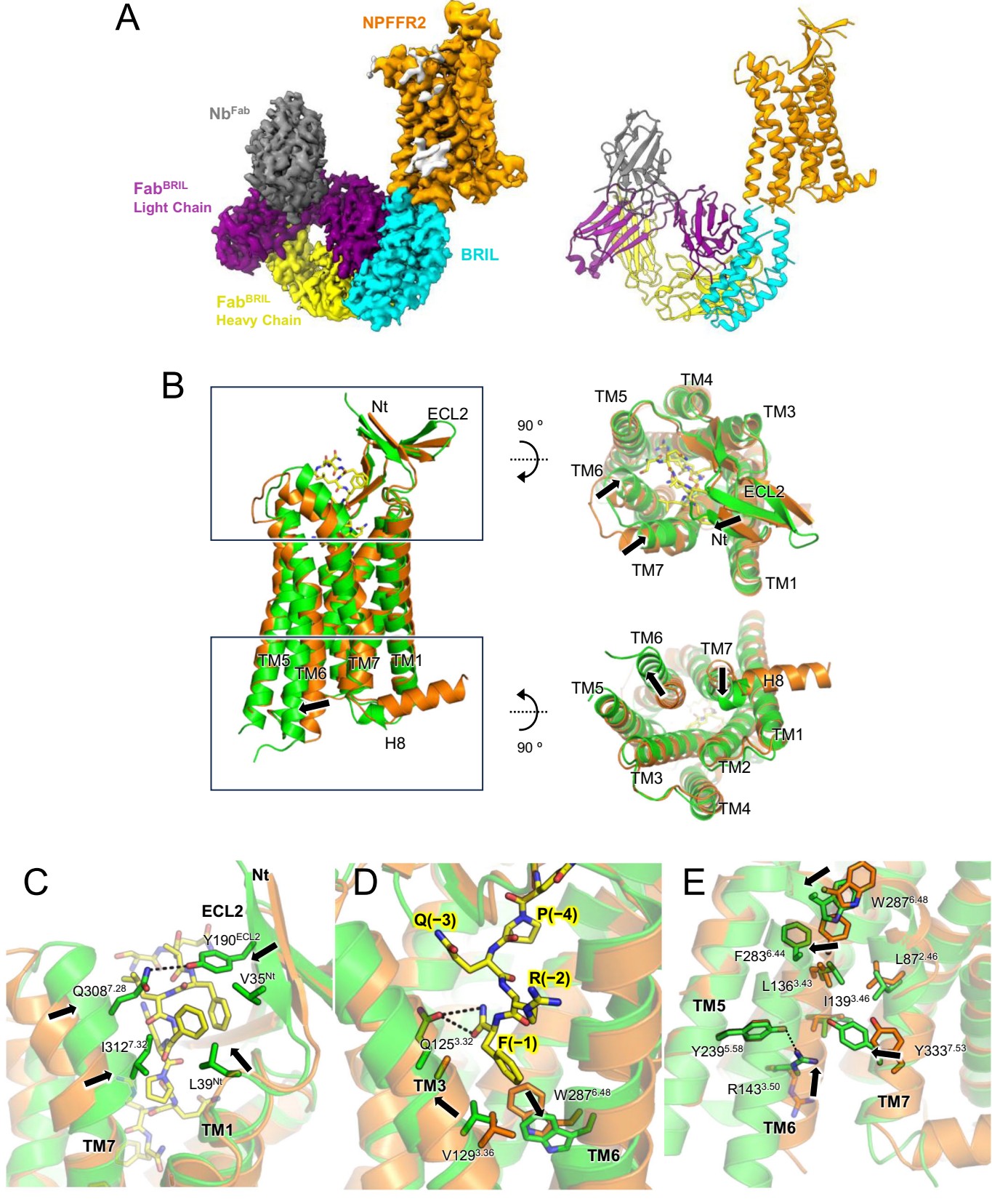

**Figure 5. Structural comparison between ligand-free and active states of NPFFR2.**

(A) The overall cryo-EM density map and structure model of the NPFFR2-BRIL–Fab–Nb complex are displayed with the following color scheme: ligand-free NPFFR2 (orange), BRIL (cyan), heavy chains of anti-BRIL Fab (Fab$^{BRIL}$) (yellow), light chains of Fab$^{BRIL}$ (purple), and anti-Fab nanobody (Nb$^{Fab}$) (gray). (B) The superposition of active (green) and ligand-free NPFFR2 (orange) structures clearly shows conformational changes upon ligand binding. The movements of TM6 and TM7 at the extracellular region, along with the shifts of ECL2 and the N-terminus (Nt) toward the ligand, are indicated by arrows. The movements of TM6 and TM7 at the intracellular region, resulting from G-protein coupling, are also depicted with arrows. (C) The conformational changes in TM7, ECL2, and the Nt that trap the ligand are illustrated with arrows. (D) The movements of the residues at the bottom of the ligand-binding pocket including toggle switch (W$^{6.48}$) and Q$^{3.32}$ are indicated by arrows. (E) The conformational changes of residues near the DRY motif and NPxxY motif upon activation are indicated by arrows. The hydrogen bonds are indicated with dashed lines.

(BRIL) based on a prior report (Tsutsumi et al, 2020). This fusion receptor was purified in a complex with anti-BRIL Fab (Fab$^{BRIL}$) and the anti-Fab nanobody (Nb$^{Fab}$) that specifically binds to the light chain of Fab$^{BRIL}$, to stabilize the complex (Fig. 5A). Cryo-EM analysis of this complex provided a global map at a resolution of 2.9 Å and a local resolution map of the receptor at 3.6 Å (Fig. EV4; Appendix Table S2).

The ligand-free structure of NPFFR2 closely resembles the inactive crystal structure of NPY2R (PDB ID: 7DDZ), with an RMSD of 1.0 Å, indicating a high degree of structural similarity (Appendix Fig. S11). The most notable feature of the ligand-free NPFFR2 structure is the configuration of ECL2. Unlike the unstructured expanded region of ECL2 observed in the inactive NPY2R, the ECL2 of NPFFR2 forms stable β-sheets together with the N-terminal tail, even in the absence of a bound ligand. However, this structure is less stable, with less-resolved side chain rotamers, compared to the ligand-bound active state. Similar to its active structure, the TM7 in NPFFR2 extends further towards the extracellular side compared to NPY2R. In addition, TM1 is positioned further away from the center of the binding pocket, likely due to the absence of an antagonist that could stabilize the TM regions in the inactive structure of NPY2R (Tang et al, 2021).

## Conformational changes of NPFFR2 upon activation

Structural comparison between the ligand-free and active states of NPFFR2 exhibits canonical overall conformational changes, such as 7 Å outward movement of the cytoplasmic segment of TM6 in the active state (Fig. 5B). In addition, inward movements of the extracellular parts of TM6 and TM7, and the ECL3 (residues 297-312) are observed upon hNPSF binding, by forming interactions of L307$^{ECL3}$, Q308$^{7.28}$, N311$^{7.31}$, and I312$^{7.32}$ with the middle segment of the peptide (Fig. 5B). At the same time, the β-strands of ECL2 and the N-terminus are also shifted towards the ligand, making a hydrophobic interaction network among V35$^{Nt}$, L39$^{Nt}$, Y190$^{ECL2}$, V203$^{ECL2}$, and the ligand residues F($-8$)$^{SF}$ and F($-6$)$^{SF}$. Upon ligand-induced rearrangement, TM6, TM7, and ECL3 move towards ECL2 and N-tail, effectively trapping the ligand (Fig. 5C). Q308$^{7.28}$ forms a hydrogen bond with Y190$^{ECL2}$ and I312$^{7.32}$ interacts with L39$^{Nt}$ which enhances the stability between ECL2 and the TM regions, securing the ligand in place and potentially facilitating ligand activation by stabilizing the receptor's active conformation. Several residues involved in hNPSF binding or forming internal interaction network in the active state, such as L39$^{Nt}$, Y190$^{ECL2}$, Q308$^{7.28}$, and I312$^{7.32}$ exhibit poorly defined side chain density in the ligand-free state map, suggesting high flexibility of these residues in the absence of agonist.

At the bottom of the ligand-binding pocket, the side chain of Q125$^{3.32}$ is well-resolved in the active state, where it forms a polar

interaction with the C-terminal amide of hNPSF. In the ligand-free state, the density of this side chain is less resolved (Appendix Fig. S12). Along with Q125$^{3.32}$, V129$^{3.36}$ is also involved in the interaction with F($-1$)$^{SF}$, inducing an upward movement of TM3 by ~2 Å (Fig. 5D). This TM3 movement triggers subsequent shifts in F283$^{6.44}$ and W287$^{6.48}$, avoiding a clash with TM3. Ultimately, these structural changes result in the opening of the cytoplasmic parts of TM5 and TM6 (Fig. 5B). In TM7, Y333$^{7.53}$ of the NPxxY motif moves inward to form a hydrophobic interaction network with L87$^{2.46}$, L136$^{3.43}$, and I139$^{3.46}$, and make polar contacts with R143$^{3.50}$ of the DRY motif and Y239$^{5.58}$ (Fig. 5E).

## Discussion

In this study, we elucidated the structural characteristics of the hNPSF–NPFFR2–G$_i$ complex, highlighting the specific recognition of RF-amide peptides by the interactions with the N-terminal tail, ECL2 and the extracellular segments of TM6 and TM7 of NPFFR2, in addition to the conserved interactions of the C-terminal RF-amide motif of the peptide with the TMD. Furthermore, a structural comparison between the ligand-free and active states of NPFFR2 provides insights into its activation mechanism upon ligand binding. Ligand binding induces inward movements of ECL2, ECL3, and the extracellular segments of TM6 and TM7, facilitating specific interactions with the N-terminal segment of the RF-amide peptide. In addition, binding of the RF-amide motif in the TMD pocket triggers TM3-mediated conformational changes, leading to the overall activation of the receptor. Human NPSF, used in our structural study as a NPFFR2 agonist, contains additional SQA residues at the N-terminus of NPFF, which enhances its potency relative to NPFF. Another RF-amide peptide, NPAF, which shares only the C-terminal four amino acids with NPFF and includes an additional 14 amino acids, also exhibits greater potency than NPFF. Unlike hNPSF, NPAF is predicted to form an α-helix, based on Alphafold3 model (Appendix Fig. S13). Given its increased potency than NPFF, the extra N-terminal helical region may be involved in receptor binding; however, further experimental validation is required.

Although NPFFR1 and NPFFR2 share high sequence similarity with each other, they exhibit different ligand selectivity towards RF-amide peptides, leading to distinct physiological functions. The exact mechanism underlying this selectivity has not been elucidated yet. Based on our structural data of the hNPSF–NPFFR2 complex and homology modeling of NPFFR1, we propose that the hydrophobicity of the ligand-binding pocket, encompassing the amino acids at the 5th and 6th positions from the C-terminus of the peptide, determines ligand selectivity (Fig. 6). Specifically, NPFFR2

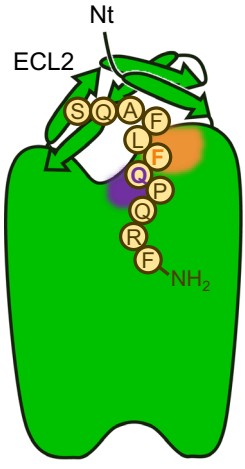
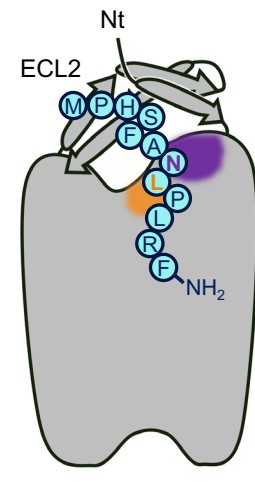

**hNPSF**-bound **NPFFR2** **GnIH**-bound NPFFR1

**Figure 6. Molecular mechanism of subtype-specific RF-amide peptide recognition by NPFFRs.**

Schematic representations of the hNPSF-bound NPFFR2 and GnIH-bound NPFFR1 are shown. While NPFFR2 and NPFFR1 share high sequence similarity, they exhibit distinct preferences for RF-amide peptide subfamilies, namely NPFFs and RFRPs. Structural comparison between the cryo-EM structure of hNPSF-bound NPFFR2 and the AlphaFold2-predicted structure of GnIH-bound NPFFR1 shows that both receptors share similar overall architecture, including a characteristic β-sheet formed by the N-terminus and ECL2, as well as conserved interactions with the C-terminal RF-amide motif of each peptide. However, a key difference is observed in the hydrophobicity of the receptor surface at the binding site for the fifth and sixth residues from the C-terminus of each peptide. This difference determines whether these NPFFRs prefer NPFFs or RFRPs.

has a hydrophobic pocket accommodating $F(-6)^{SF}$ and polar residues nearby $Q(-5)^{SF}$ of hNPSF. In contrast, NPFFR1's ligand-binding pocket consists of polar and nonpolar residues accommodating $N(-6)^{GnIH}$ and $L(-5)^{GnIH}$, respectively. This proposal is supported by the reversed selectivity observed in signaling assays using NPFFR1 mutant, where NPFFR2 residues Ler, Ser, and Asn were introduced at positions interacting with the amino acids in the $5^{th}$ and $6^{th}$ positions from the C-terminus of the peptide.

With the NPFFR2 structure, there is now an opportunity to develop drugs with significantly better selectivity than previously available compounds. Prior multitarget ligands, such as BN-9 and DN-9, demonstrated efficacy by acting on both opioid receptors and NPFFR2, but lacked precise selectivity between NPFFR2 and NPFFR1 (Li et al, 2016; Xu et al, 2020). These drugs offered analgesic effects while mitigating opioid-related side effects like tolerance and hyperalgesia, yet their cross-reactivity with NPFFR1 posed challenges in fully optimizing their therapeutic potential. The detailed structural understanding of NPFFR2 in its inactive and active states now opens the door to designing drugs with enhanced receptor subtype selectivity. This knowledge facilitates the development of ligands with enhanced specificity for targeting NPFFR2, reducing the off-target effects associated with NPFFR1 interaction. Such specificity could lead to bifunctional drugs that not only improve pain relief but also further limit opioid-induced side effects like constipation and dependence, making them more effective and safer for long-term use.

Further research is needed to fully comprehend the distinct activation mechanisms of these receptors, especially given the observed differences in how they respond to various ligands, including recent reports that NPFFRs also act as receptors for non-RF-amide peptides such as kissorphin (Milton, 2012). Expanding this knowledge will enhance our understanding of their roles in physiological processes and inform the development of targeted therapeutic strategies.

# Methods

### Reagents and tools table

| Reagent/resource | Reference or source | Identifier or catalog number |
|---|---|---|
| **Experimental models** | | |
| *E. coli* ROSETTA(DE3) | Novagen | Cat# 70954 |
| *Spodoptera frugiperda* (Sf9) | Expression systems | Cat#94-001F |
| High Five (BTI-Tn-5B1-4) | Expression systems | Cat#94-002F |
| HEK293T | ATCC | Cat#CRL-3216 |
| **Recombinant DNA** | | |
| pFastBac Dual anti-BRIL Fab | This study | N/A |
| pcDNA3.1 NPFFR2 | This study | N/A |
| pcDNA3.1 NPFFR1 | This study | N/A |
| pFastBac NPFFR2 | This study | N/A |
| pFastBac NPFFR2-BRIL | This study | N/A |
| pFastBac scFv16 | This study | N/A |
| pFastBac Gα$_{i1}$ | This study | N/A |
| pFastBac Dual Gβ$_1$, and Gγ$_2$ | This study | N/A |
| pET-21d anti-Fab Nanobody | This study | N/A |
| GloSensor plasmid | Promega | Cat#E2301 |
| **Antibodies** | | |
| Rabbit anti-FLAG antibody | Cell Signaling Technology | Cat#14793 |
| Anti-rabbit HRP-conjugated antibody | Enzo Life Sciences | Cat#ADI-SAB-300 |
| **Oligonucleotides and other sequence-based reagents** | | |
| Peptides | Cusabio | Custom order |
| **Chemicals, enzymes, and other reagents** | | |
| ESF921 incest cell culture medium | Expression system | Cat#96-001-20 |
| Cellfectin™ II Reagent | Gibco | Cat#10352100 |
| Lauryl Maltose Neopentyl Glycol | Anatrace | Cat#NG310 |
| Cholesteryl hemisuccinate | Sigma-Aldrich | Cat#C6512 |
| n-Dodecyl-β-D-Maltopyranoside | Anatrace | Cat#D310A |
| Leupeptin | Goldbio | Cat#L-010-5 |
| Benzamidine | Sigma-Aldrich | Cat#B6506 |
| Phenylmethylsulfonyl fluoride | Sigma-Aldrich | Cat#11359061001 |
| Tris(2-carboxyethyl) phosphine hydrochloride | Goldbio | Cat#TCEP1 |

| Reagent/resource | Reference or source | Identifier or catalog number |
|---|---|---|
| Apyrase | NEB | Cat#M0398L |
| Dulbecco's Modified Eagle Medium (DMEM) | Cytiva | Cat#SH30243.01 |
| Fetal Bovine Serum | GWvitek | Cat#US-FBS-500 |
| Antibiotic-Antimycotic | Gibco | Cat#15240-062 |
| Lipofectamine 2000 | Invitrogen | Cat#11668019 |
| 4% Paraformaldehyde | Tech&Innovation | Cat#BPP-9004 |
| BSA | Bovogen Biologicals | Cat#BSAS 0.1 |
| 1-Step™ TMB-Blotting Substrate Solution | Thermofisher Scientific | Cat#34018 |
| Janus Green B | Tokyo Chemical Industry | Cat#J0002 |
| Forskolin | Sigma-Aldrich | Cat#F6886 |
| $CO_2$-independent medium | Gibco | Cat#15420604 |
| D-Luciferin | NanoLight | Cat#306 |
| Accutase solution | Sigma-Aldrich | Cat# A6964 |
| Software | | |
| CryoSPARC | https://cryosparc.com/ | Version 4.5.1 |
| PHENIX | https://phenix-online.org/ | Version 1.20.1-4487 |
| COOT | www.2.mrc-lmb.cam.ac.uk/personal/pemsley/coot/ | Version 0.9.8.1 |
| UCSF ChimeraX | https://www.cgl.ucsf.edu/chimerax/ | Version 1.6.1 |
| GraphPad Prism | https://www.graphpad.com/scientific-software/prism/ | Version 10.1.2 |
| MolCube PDB Reader | https://www.molcube.xyz/software | |
| Membrane Builder | CHARMM-GUI | |
| GROMACS | https://www.gromacs.org/ | |
| CHARMM36(m) | CHARMM-GUI | |
| AlphaFold2 | https://colab.research.google.com/github/sokrypton/ColabFold/blob/main/AlphaFold2.ipynb#scrollTo=kOblAo-xetgx | ColabFold v1.5.5 |
| AlphaFold3 | https://alphafoldserver.com/ | |
| Cuemol2 | http://www.cuemol.org/en/ | Version 2.2.3.443 |
| Other | | |
| FlexStation 3 multi-mode microplate reader | Molecular Devices | |

## Expression and purification of NPFFR2 and NPFFR2-BRIL

The human NPFFR2 (8-420) was modified to include affinity tags (an N-terminal FLAG tag and a C-terminal GFP followed by 8xHis tag) for purification purposes. The NPFFR2-BRIL construct was designed to include cytochrome b562 RIL (BRIL) protein replacing residues 246-266 in the ICL3 region, serving as a fiducial marker for cryo-EM study. Both NPFFR2 and NPFFR2-BRIL were expressed in *Spodoptera frugiperda* (Sf9) cells using the Bac-to-Bac system (Invitrogen). Forty-eight hours post-infection, cells were harvested by centrifugation at 4000 rpm for 10 min. The harvested cells were lysed with a Dounce homogenizer in lysis buffer (20 mM Tris-Cl, pH 8.0, 150 mM NaCl) containing protease inhibitors. Membrane fractions were collected by centrifugation at 14,000 rpm for 30 min at 4 °C, resuspended in solubilization buffer (20 mM Tris-Cl, pH 8.0, 150 mM NaCl, 1% (w/v) lauryl maltose neopentyl glycol (LMNG), 0.1% (w/v) cholesterol hemisuccinate (CHS), and protease inhibitors), and homogenized with a Dounce homogenizer before being stirred for 2 h at 4 °C. After solubilization, the mixture was centrifuged at 14,000 rpm for 30 min at 4 °C to obtain the supernatant. The supernatant containing solubilized NPFFR2 or NPFFR2-BRIL was then purified using a Ni-NTA affinity column. The column was washed with wash buffer (20 mM Tris-Cl, pH 8.0, 150 mM NaCl, 20 mM Imidazole, 0.01% (w/v) LMNG, 0.001% (w/v) CHS) and the bound protein was eluted with elution buffer containing 20 mM Tris-Cl, pH 8.0, 150 mM NaCl, 250 mM Imidazole, 0.01% (w/v) LMNG, and 0.001% (w/v) CHS.

## Expression and purification of heterotrimeric $G_i$

The human $G\alpha_{i1}$, 6xHis-$G\beta_1$, and $G\gamma_2$, were co-expressed in BTI-Tn-5B1-4 (High Five) cells. After 48 h of infection, cells were collected and lysed using lysis buffer (20 mM Tris-Cl, pH 8.0, with protease inhibitors). The cell lysate was subjected to centrifugation at 14,000 rpm for 20 min at 4 °C to separate the insoluble material. The G-protein complex was then solubilized in a buffer containing 20 mM Tris-Cl, pH 8.0, 100 mM NaCl, 2 mM $MgCl_2$, 50 µM GDP, 1% (w/v) n-Dodecyl β-D-Maltoside (DDM), 0.1 mM TCEP, and protease inhibitors. The mixture was centrifuged again to remove any insoluble debris form the lysate, and the supernatant was applied to a Ni-NTA affinity column. The protein was eluted using a buffer composed of 20 mM Tris-Cl, pH 8.0, 100 mM NaCl, 1 mM $MgCl_2$, 10 µM GDP, 0.03% (w/v) DDM, 0.1 mM TCEP, and 300 mM imidazole. The eluted protein was further purified by ion-exchange chromatography on a HiTrap Q column (Cytiva) equilibrated with 20 mM Tris-Cl, pH 8.0, 1 mM $MgCl_2$, 10 µM GDP, and 0.03% (w/v) DDM. The fractions containing $G_i$ were pooled, concentrated, flash-frozen in liquid nitrogen, and stored at −80 °C until use.

## Expression and purification of scFv16

The single-chain variable fragment scFv16, which stabilizes the $G_i$ protein, was purified following a modified version of a previously described protocol (Maeda et al, 2018). Briefly, High Five cells were used to express scFv16 with a C-terminal 8xHis tag. After harvesting the cells, the culture supernatant containing the secreted scFv16 was incubated with Ni-NTA resin for 2 h at 4 °C. The resin was then washed with a buffer containing 20 mM Tris-Cl, pH 7.5, 150 mM NaCl, and 30 mM imidazole. The bound scFv16 was eluted with the same buffer containing 300 mM imidazole. The eluted protein was further purified by size exclusion chromatography using a Superdex 200 10/300 gel filtration column (Cytiva), which was pre-equilibrated with 20 mM HEPES, pH 7.0, and 150 mM NaCl. The purified scFv16 was concentrated, flash-frozen in liquid nitrogen, and stored at −80 °C until further use.

## Purification of the hNPSF–NPFFR2–$G_i$–scFv16 complex

To generate the NPFFR2–$G_i$ complex, the purified proteins, NPFFR2, $G_i$ heterotrimer, and scFv16 were mixed in a molar ratio of 1:1.5:2 in the presence of the synthesized hNPSF ligand (0.01 mM). This mixture was incubated overnight at 4 °C with the addition of HRV3C protease (prepared in-house) to cleave the GFP tag and 5 units of apyrase (NEB) to remove GDP. Following

incubation, the protein mixture was applied to an M1 anti-FLAG affinity column (prepared in-house) to separate the desired complex from excess $G_i$ and scFv16. The bound complex was eluted using FLAG peptide. Throughout the purification process, the samples were maintained in 0.01% (w/v) LMNG and 0.001% (w/v) CHS during the washing steps. The final purification was performed by size exclusion chromatography using a buffer consisting of 20 mM Tris-Cl, pH 8.0, 150 mM NaCl, 0.001% (w/v) LMNG, 0.0001% (w/v) CHS, and 100 nM hNPSF. The purified hNPSF–NPFFR2–$G_i$–scFv16 complex was concentrated to 4 mg/ml and used for grid preparation.

## Expression and purification of anti-BRIL Fab

The genes encoding VH and VL domain of the BRIL-specific Fab (Fab$^{BRIL}$) were cloned into the pFastBac Dual vector for expression in High Five cells using the Bac-to-Bac system (Tsutsumi et al, 2020). Seventy-two hours post-infection, the cells were harvested, and the supernatant was collected. This supernatant was then applied to a Ni-NTA resin column. After washing, the Fab was eluted with 20 mM Tris-Cl, pH 8.0, 150 mM NaCl, 250 mM imidazole. The eluted Fab was concentrated and further purified using a HiLoad 26/200 Superdex 200 column (Cytiva) with a buffer containing 20 mM HEPES, pH 7.5, and 200 mM NaCl. The purified Fab$^{BRIL}$ was then concentrated, flash-frozen in liquid nitrogen, and stored at −80 °C until further use.

## Expression and purification of nanobody for BRIL–Fab

The nanobody recognizing the light chain of BRIL–Fab (Nb$^{Fab}$) was cloned into pET-21d E.coli expression vector with a C-terminal 6xHis-tag. Expression was carried out in Rosetta (DE3, Novagen) strain, and Nb$^{Fab}$ was purified as previously described (Tsutsumi et al, 2020). Briefly, Nb$^{Fab}$ was first purified using a Ni-NTA column and then further purified by size exclusion chromatography with a Superdex 200 10/300 column (Cytiva), sequentially. The peak fractions were collected and stored at 4 °C until use.

## Purification of the NPFFR2-BRIL–Fab$^{BRIL}$–Nb$^{Fab}$ complex

The purified NPFFR2-BRIL, Fab$^{BRIL}$, and Nb$^{Fab}$ were mixed at a molar ratio of 1:1.2:1.5 with HRV3C protease to cleave the C-terminal GFP tag from NPFFR2-BRIL. The mixture was incubated overnight at 4 °C. Subsequently, the complex was purified using a Superdex 200 10/300 column (Cytiva) pre-equilibrated with a buffer containing 20 mM Tris-Cl, pH 8.0, 150 mM NaCl, 0.001% (w/v) LMNG, and 0.0001% (w/v) CHS, to remove excess Fab$^{BRIL}$ and Nb$^{Fab}$, as well as GFP and HRV3C protease. The peak fractions were collected and concentrated to 1.5 mg/ml for preparing cryo-EM grids.

## Cryo-EM grid preparation and data collection

The purified hNPSF–NPFFR2–$G_i$–scFv16 complex and NPFFR2-BRIL–Fab$^{BRIL}$–Nb$^{Fab}$ complex, containing 0.01% digitonin, were applied onto freshly glow-discharged QUANTIFOIL® R holey carbon grids (R0.6/1.0, 300 mesh; SPI) that had been coated with poly-L-lysine. The grids were then plunge-frozen in liquid ethane using a Vitrobot Mark IV (Thermo Fisher Scientific, SNU CMCI) and prescreened on a 200 kV Glacios TEM (Thermo Fisher Scientific, SNU CMCI) equipped with a Falcon 4 detector using EPU software (Thermo Fisher Scientific). Data collection for the hNPSF–NPFFR2–$G_i$–scFv16 complex was performed on a Titan Krios microscope (Thermo Fisher Scientific) equipped with a Falcon 4i camera at Pusan National University, Korea. A total of 22,278 movies were recorded with a calibrated pixel size of 1.05 Å, using a defocus range between −0.8 and −2.0 μm. Each movie was captured over 5.88 s, divided into 40 frames, resulting in a cumulative electron dose of ~45 e/Å². Data collection for the NPFFR2-BRIL–Fab$^{BRIL}$–Nb$^{Fab}$ complex was conducted on a Titan Krios G4 microscope (Thermo Fisher Scientific) equipped with a BioQuantum K3 detector at the Institute of Membrane Proteins (Pohang, Korea). For the NPFFR2-BRIL–Fab$^{BRIL}$–Nb$^{Fab}$ complex, 14,597 movies were recorded with a pixel size of 1.05 Å and a defocus range of −0.8 to −2.0 μm. These movies were captured over 4.77 s and divided into 50 frames, with a total electron dose of around 60 e/Å².

## Data processing and 3D map reconstruction

All movies were dose-fractionated and corrected for beam-induced motion using the patch motion correction algorithm in cryoSPARC v4.5.1 (Punjani et al, 2017). Contrast Transfer Function (CTF) parameters were estimated with patch CTF estimation in cryoSPARC. Initial particle picking was performed using the blob picker tool in cryoSPARC, followed by 2D classification to identify the best particles. These particles were then used to train Topaz, which was employed for improved particle picking.

For the hNPSF–NPFFR2–$G_i$–scFv16 complex, a total of 6,152,021 particles were picked using Topaz (Bepler et al, 2019). These particles underwent two rounds of 2D classification, resulting in a refined set of particles. This refined set was then used for ab initio reconstruction and heterogeneous refinement, yielding 990,464 particles for 3D classification. NPFFR2 mask was created for focused 3D classification, and the selected particles from each classification were combined for a final refinement.

For the NPFFR2-BRIL–Fab$^{BRIL}$–Nb$^{Fab}$ complex, a total of 6,231,249 particles were picked using Topaz. These particles underwent one round of 2D classification, resulting in a refined set of particles. This refined set was then used for ab initio reconstruction and heterogeneous refinement, yielding 1,717,987 particles for 3D classification. A similar workflow was followed, with 3D classification using a NPFFR2 mask. Local CTF correction was performed for the NPFFR2-BRIL–Fab$^{BRIL}$–Nb$^{Fab}$ complex.

Non-uniform (NU) refinement was performed in cryoSPARC using default settings, yielding the final maps with global nominal resolutions of 3.2 Å for the hNPSF–NPFFR2–$G_i$ complex and 2.9 Å for the NPFFR2-BRIL–Fab$^{BRIL}$–Nb$^{Fab}$ complex, as determined by the gold standard Fourier Shell Correlation (GSFSC) at the 0.143 criterion. To facilitate model building, local refinements were performed separately for NPFFR2, the $G_i$–scFv16 complex, and the BRIL–Fab$^{BRIL}$–Nb$^{Fab}$ complex using individual masks. These refinements employed pose/shift Gaussian prior during alignment, with a rotation standard deviation of 5 degrees and a shift standard deviation of 2 Å. For the hNPSF–NPFFR2–$G_i$–scFv16 complex, the local resolution of NPFFR2 is 3.4 Å, while the G-protein–scFv16 region reaches 3.0 Å. In the NPFFR2-BRIL–Fab$^{BRIL}$–Nb$^{Fab}$ complex, the local resolution of NPFFR2 is 3.6 Å, and the BRIL–Fab$^{BRIL}$–Nb$^{Fab}$ region achieves 2.8 Å. The locally refined maps were integrated using the 'vop maximum' function to generate each composite map, which was used for representation in Figs. 1B and 5A.

## Model building and refinement

The model building and refinement were conducted based on the locally refined sharp maps. Model building was carried out manually using ChimeraX v.1.6.1 (Pettersen et al, 2021) and COOT v.0.9.8.1 (Emsley and Cowtan, 2004). For the hNPSF–NPFFR2–$G_i$–scFv16 complex, the NPFFR2 structure was modeled using an AlphaFold2 prediction as the initial template, while $G_i$–scFv16 structure was modeled based on PDB entry 7YON (NPY–NPY2R–$G_i$–scFv16). For the NPFFR2-BRIL–Fab[BRIL]–Nb[Fab] complex, the AlphaFold2-predicted model was used as the initial template, while the BRIL–Fab[BRIL]–Nb[Fab] structure was derived from PDB entry 7TUY. Each model was further refined using the real-space refinement in Phenix v.1.20.1 (Liebschner et al, 2019), ensuring accurate fitting to the density map. Detailed cryo-EM model refinement statistics, analyzed using Phenix validation based on global maps, are provided in Appendix Table S2.

## GloSensor™ cAMP signaling assay

HEK293T cells were seeded at a density of $6 \times 10^5$ cells per well in a six-well cell culture plate (SPL) and co-transfected with expression plasmids encoding GloSensor™ and each NPFFR construct, using 1 μg to 4 μg of DNA per well according to the Lipofectamine™ 2000 Transfection Reagent (Invitrogen) protocol. After 48 h of incubation under standard conditions (37 °C, 5% $CO_2$), the cells were detached using Accutase solution (Sigma-Aldrich) and washed with PBS. The cells were then pelleted by centrifugation, resuspended in $CO_2$-independent medium (Gibco) supplemented with Firefly D-Luciferin (0.5 mg/ml, NanoLight), and distributed into a 96-well white plate (SPL). After a 2-hour incubation, serially diluted ligands and 0.5 μM forskolin were added to each well to detect ligand-induced signaling. Luminescence signals were measured using a Mithras LB940 instrument (Berthold) with a measurement time of 1 s per well.

## ELISA-based surface expression assay

HEK293T cells were seeded in a 96-well cell culture plate (SPL) and transfected with expression plasmids encoding NPFFR wild-type and mutants, following the Lipofectamine™ 2000 Transfection Reagent (Invitrogen) protocol. After 48 h of incubation under standard conditions (37 °C, 5% $CO_2$), the cells were fixed with 4% paraformaldehyde (PFA, T&I) at room temperature for 10 min and immediately washed with PBS. Blocking was performed with 5% bovine serum albumin (BSA, Bovogen) for 2 h, followed by staining with rabbit anti-FLAG antibody (Cell Signaling Technology, 1:1000 dilution). Detection was carried out using an anti-rabbit HRP antibody (Enzo Life Science, 1:1000 dilution). After 2 h, TMB solution (Thermo Fisher Scientific) was added and incubated for 20 min. The reaction was quenched with 1 M HCl, and absorbance was measured at 450 nm using a FlexStation 3 multi-mode microplate reader (Molecular Devices). Each well was then stained with Janus Green solution (0.2% (w/v), TCL) and washed with Milli-Q water. After adding 0.1 M HCl, absorbance was measured at 595 nm. The target receptor expression level was normalized by calculating the ratio of the absorbance ($Abs_{450}/Abs_{595}$). Graphs were plotted using GraphPad Prism 10.1.2.

## All-atom MD simulations

An all-atom MD simulation system of the hNPSF–NPFFR2 complex was prepared for all-atom MD simulation with dimensions $90.3 \times 90.3 \times 143.0$ Å$^3$ and comprising 111,248 atoms (77,089 water molecules). In the final model of the hNPSF-bound NPFFR2 structure, due to poor EM map quality, the N-terminal 24 residues and residues beyond 348 were excluded.

The hNPSF-bound NPFFR2 model was embedded into a model membrane composed of 1-palmitoyl-2-oleoyl-sn-glycero-3-phosphocholine (POPC) and cholesterol in a ratio of 4:1. MolCube PDB Reader and Membrane Builder, a commercial software similar to CHARMM-GUI PDB Reader and Membrane Builder, was used for this purpose (Jo et al, 2008; Jo et al, 2009; Park et al, 2023). The CHARMM36(m) force field (Hatcher et al, 2009; Huang et al, 2017) was utilized for lipids and proteins. The NPFFR2 protein N-terminus was acetylated and the C-terminus was methylamidated to remain neutral. The N-terminus of the hNPSF was modeled with the standard N-terminal group, and the C-terminus was amidated. For solvation, 0.15 M NaCl was included with the TIP3P water model (Jorgensen et al, 1983).

Van der Waals interactions were smoothly switched off between 10 and 12 Å using a force-switch modifier (Dion et al, 2004), long-range electrostatic interactions were calculated using the Particle-Mesh Ewald (PME) electrostatics method (Essmann et al, 1995) with a cutoff of 12 Å. The LINCS algorithm was used to constrain bond lengths, including hydrogen atoms. Simulations were conducted at 310 K and 1 bar, respectively (Ryckaert et al, 1977). Following the Membrane-Builder's six-step equilibration procedure (Jo et al, 2007; Lee et al, 2016), NVT (constant particle number, volume, and temperature) simulations with positional and dihedral restraints were performed to gradually alleviate the force constants. Subsequently, NPT (constant particle number, pressure, and temperature) simulations were conducted for production runs without any restraints, using a 4 fs time-step with the hydrogen mass repartitioning method (Gao et al, 2021; Hopkins et al, 2015). The Nosé-Hoover and Parrinello-Rahman methods were used to maintain constant temperature and pressure, with coupling constants set to 1.0 ps and 5.0 ps, respectively (Hoover, 1985; Nose, 1984). Periodic boundary conditions were applied for all simulations, with five independent simulations of the hNPSF–NPFFR2 system to ensure adequate sampling. All simulations were carried out using GROMACS was used for 1.5 μs each (Pronk et al, 2013).

## Statistical analysis

Data from the GloSensor™ cAMP signaling assays were analyzed using GraphPad Prism with a three-parameter logistic model to fit concentration-response curves for both wild-type (WT) and mutant receptors. Mutant expression levels and DNA amounts were adjusted to match WT conditions. $\Delta pEC_{50}$ values were calculated by subtracting the log $EC_{50}$ of the WT receptor from that of each mutant. Results are presented as mean ± SEM from three independent biological replicates, each performed in duplicate or more. Error bars represent the SEM. Statistical significance was assessed using one-way ANOVA, followed by Dunnett's tests comparisons to the WT receptor. Significance levels were reported as follows: ns (not significant, $P > 0.05$); *$P < 0.05$; **$P < 0.01$; ***$P < 0.001$.

## Data availability

The cryo-EM density maps and corresponding atomic coordinates for the hNPSF–NPFFR2–G$_i$–scFv16 complex have been deposited in the Electron Microscopy Data Bank (EMDB) under accession code EMD-61444 (https://www.ebi.ac.uk/emdb/EMD-61444) and in the Protein Data Bank (PDB) under accession code 9JFY (https://www.rcsb.org/structure/9JFY). The cryo-EM density maps and atomic coordinates for the NPFFR2–BRIL–Fab$^{BRIL}$–Nb$^{Fab}$ complex have been deposited in the EMDB under accession code EMD-61446 (https://www.ebi.ac.uk/emdb/EMD-61446) and in the PDB under accession code 9JG0 (https://www.rcsb.org/structure/9JG0). The sharpened composite maps generated by combining the locally refined maps have been deposited in the EMDB as additional maps. All data are publicly available and can be accessed online.

The source data of this paper are collected in the following database record: biostudies:S-SCDT-10_1038-S44319-025-00428-2.

## Peer review information

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

## Acknowledgements

This work was supported by the Global-LAMP Program of the National Research Foundation of Korea (NRF) grant funded by the Ministry of Education (No. RS-2023-00301976 to Jeesoo K) and by the Tech Incubator Program for Startup (S3319679 to WI) funded by the Ministry of SMEs and Startups (MSS, Korea). In addition, this work was supported by the Bio&Medical Technology Development Program (No. RS-2024-00344154 to H-JC) and the NRF grants (No. 2023R1A2C3004205 and RS-2024-00407331 to H-JC) funded by the Korean government (MSIT). We thank the cryo-EM facilities of NEXUS consortium, supported by a NRF of Korea grant RS-2024-00440289 (to H-JC). We thank the Global Science Experimental Data Hub Center (GSDC) and KREONET at the Korea Institute of Science and Technology Information (KISTI) for computing resources and technical support.

## Author contributions

**Jeesoo Kim**: Data curation; Investigation; Visualization; Writing—original draft. **Sooyoung Hong**: Data curation; Investigation. **Hajin Lee**: Data curation. **Hyun Sik Lee**: Data curation. **Chaehee Park**: Validation; Visualization; Writing—original draft. **Jinuk Kim**: Validation. **Wonpil Im**: Formal analysis; Supervision. **Hee-Jung Choi**: Conceptualization; Supervision; Writing—original draft; Writing—review and editing.

Source data underlying figure panels in this paper may have individual authorship assigned. Where available, figure panel/source data authorship is listed in the following database record: biostudies:S-SCDT-10_1038-S44319-025-00428-2.

## Disclosure and competing interests statement

The authors declare no competing interests.

# Expanded View Figures

**Figure EV1. Sample preparation and cryo-EM data analysis of the hNPSF–NPFFR2–G$_i$ complex.**

(A, B) Size exclusion chromatography and SDS-PAGE profile of hNPSF–NPFFR2–G$_i$ complex used in the cryo-EM sample preparation. (C) Representative image of cryo-EM micrograph (scale bar: 140 nm) and 2D classification results. (D) Cryo-EM data processing workflow using CryoSPARC. (E) Gold standard Fourier shell correlation (GSFSC) curve and direction distribution profile. (F) Representative image of electron density map and model of the hNPSF–NPFFR2–G$_i$ complex. Source data are available online for this figure.

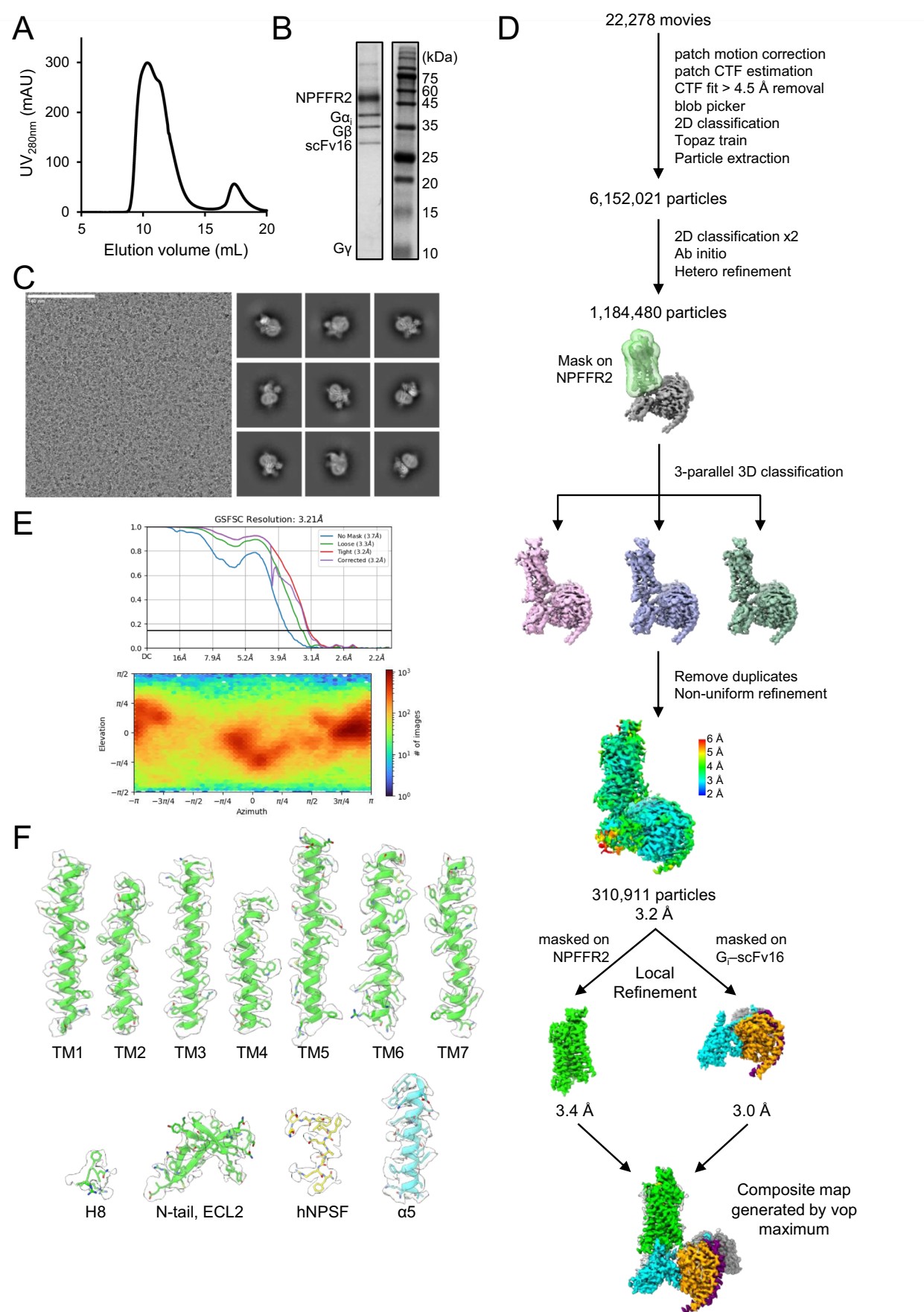

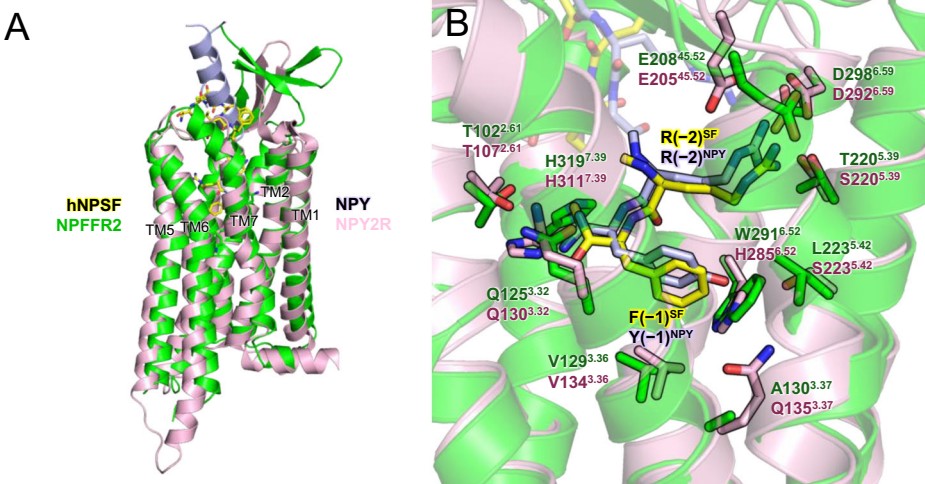

**Figure EV2. Comparison of the agonist binding pockets of NPFFR2 and NPY2R.**

(A) Superposition of the hNPSF–NPFFR2–G$_i$ complex and the NPY–NPY2R–G$_i$ complex (PDB: 7YOO). hNPSF is colored yellow, NPFFR2 is green, NPY is light purple, and NPY2R is pink. The structures are aligned based on the receptors, with an overall RMSD of 0.9 Å. (B) A detailed view at the bottom of the ligand-binding pocket of NPFFR2 and NPY2R reveals that two C-terminal residues of hNPSF and NPY share similar binding poses. NPY2R residues involved in the interaction toward Arg-Tyr-NH$_2$ and the corresponding residues in NPFFR2 are shown in sticks.

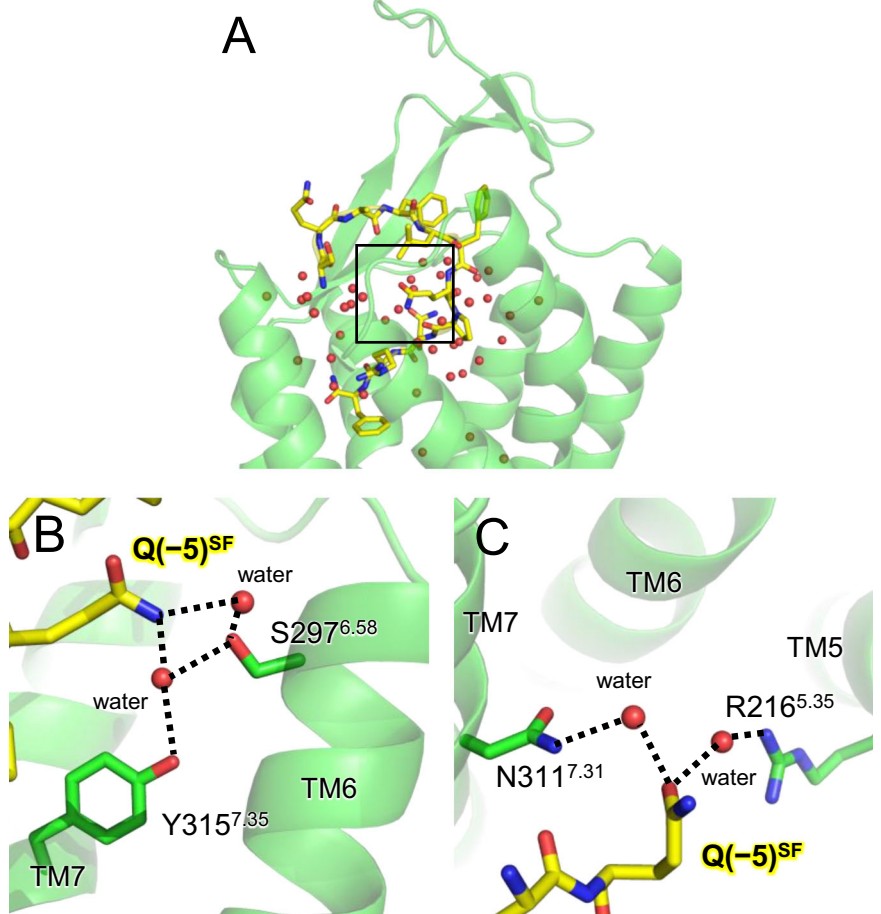

**Figure EV3. Water-mediated interaction between hNPSF Q(−5) and NPFFR2.**

(A) Water molecules near the extracellular regions of TMD are shown as red spheres in the all-atom MD simulation. (B, C) Water-mediated interactions involving Q(−5)[SF], based on model structures from MD simulation frames. The ligand and receptor residues (R216[5.35], S297[6.58], N311[7.31] and Y315[7.35]) form hydrogen bonds with water molecules, with N and O atoms positioned within 3.5 Å.

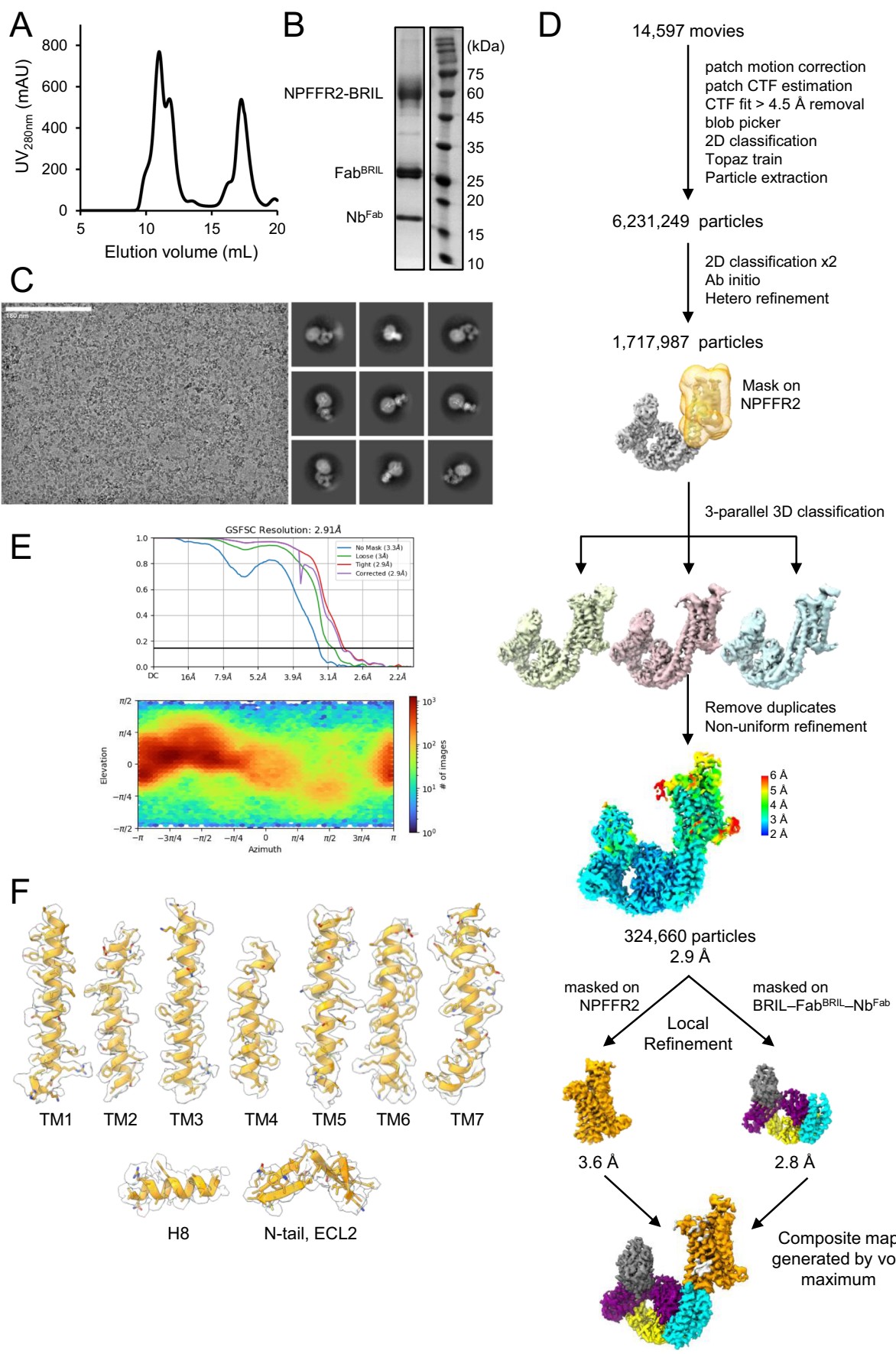

◀ **Figure EV4. Sample preparation and cryo-EM data analysis of the NPFFR2-BRIL–Fab<sup>BRIL</sup> –Nb<sup>Fab</sup> complex.**

(A, B) Size exclusion chromatography and SDS-PAGE profile of the NPFFR2-BRIL–Fab<sup>BRIL</sup>–Nb<sup>Fab</sup> complex used in the cryo-EM sample preparation. (C) Representative image of cryo-EM micrograph (scale bar: 180 nm) and 2D classification results. (D) Cryo-EM data processing workflow using CryoSPARC. (E) Gold standard Fourier shell correlation (GSFSC) curve and direction distribution profile. (F) Representative image of electron density map and model of the NPFFR2-BRIL–Fab<sup>BRIL</sup>–Nb<sup>Fab</sup> complex. Source data are available online for this figure.

