## [Peer Review File · EMBO Reports]

Structural insights into the selective recognition of RF-amide peptides by neuropeptide FF receptor 2

Jeesoo Kim, Sooyoung Hong, Hajin Lee, Hyun Sik Lee, Chaehee Park, Jinuk Kim, Wonpil Im, and Hee-Jung Choi

Corresponding author(s): Hee-Jung Choi (choihj@snu.ac.kr)

Review Timeline:

Submission Date:	24th Oct 24
Editorial Decision:	10th Dec 24
Revision Received:	24th Jan 25
Editorial Decision:	26th Feb 25
Revision Received:	6th Mar 25
Accepted:	11th Mar 25

Editor: Esther Schnapp

Transaction Report:

Dear Prof. Choi,

Thank you for the submission of your manuscript to EMBO reports. We have now received the enclosed comments from 2 referees and given that both are in fair agreement, I am making a decision on your ms now in order to save time.

As you will see, the referees acknowledge that the findings are interesting. However, they also have several suggestions for how the manuscript could be improved and strengthened, and I think all suggestions are good and should be addressed. Please let me know if you disagree and we can discuss the exact revision requirements further, also in a video chat, if you like.

I would thus like to invite you to revise your manuscript with the understanding that the referee concerns must be fully addressed and their suggestions taken on board. Please address all referee concerns in a complete point-by-point response. Acceptance of the manuscript will depend on a positive outcome of a second round of review. It is EMBO reports policy to allow a single round of major revision only and acceptance or rejection of the manuscript will therefore depend on the completeness of your responses included in the next, final version of the manuscript.

We realize that it is difficult to revise to a specific deadline. In the interest of protecting the conceptual advance provided by the work, we recommend a revision within 3 months (12th Mar 2025). Please discuss the revision progress ahead of this time with the editor if you require more time to complete the revisions.

- 1) A data availability section providing access to data deposited in public databases is missing. If you have not deposited any data, please add a sentence to the data availability section that explains that.
- 2) Your manuscript contains statistics and error bars based on $n=2$. Please use scatter blots in these cases. No statistics should be calculated if $n=2$.

5) a complete author checklist, which you can download from our author guidelines <https://www.embopress.org/page/journal/14693178/authorguide>. Please insert information in the checklist that is also reflected in the manuscript. The completed author checklist will also be part of the RPF.

6) Please note that all corresponding authors are required to supply an ORCID ID for their name upon submission of a revised manuscript (<https://orcid.org/>). Please find instructions on how to link your ORCID ID to your account in our manuscript tracking system in our Author guidelines <https://www.embopress.org/page/journal/14693178/authorguide#authorshipguidelines>

7) Before submitting your revision, primary datasets produced in this study need to be deposited in an appropriate public

database (see <https://www.embopress.org/page/journal/14693178/authorguide#datadeposition>). Please remember to provide a reviewer password if the datasets are not yet public. The accession numbers and database should be listed in a formal "Data Availability" section placed after Materials & Method (see also <https://www.embopress.org/page/journal/14693178/authorguide#datadeposition>). Please note that the Data Availability Section is restricted to new primary data that are part of this study. * Note - All links should resolve to a page where the data can be accessed. *

10) Regarding data quantification (see Figure Legends:

<https://www.embopress.org/page/journal/14693178/authorguide#figureformat>)

12) All Materials and Methods need to be described in the main text using our 'Structured Methods' format, which is required for all research articles. According to this format, the Methods section includes a separate Reagents and Tools Table file (listing key reagents, experimental models, software and relevant equipment and including their sources and relevant identifiers) and a Methods and Protocols section describing the methods using a step-by-step protocol format. The aim is to facilitate adoption of the methodologies across labs. More information on how to adhere to this format as well as a downloadable template (.docx) for the Reagents and Tools Table can be found in our author guidelines:

An example of a Method paper with Structured Methods can be found here: <https://www.embopress.org/doi/full/10.1038/s44320-024-00037-6#sec-4>

You are able to opt out of this by letting the editorial office know (emboreports@embo.org). If you do opt out, the Review

Process File link will point to the following statement: "No Review Process File is available with this article, as the authors have chosen not to make the review process public in this case."

I look forward to seeing a revised form of your manuscript when it is ready.

Yours sincerely,

Referee #1:

Kim et al have determined the cryo-EM structure of an active state of the neuropeptide FF receptor 2 (NPFFR2) bound to the agonist peptide hNDSF and coupled to heterotrimeric Gi, and also the cryo-EM structure of an inactive ligand-free state. The authors use the structure to identify regions that define the specificity of peptide binding, in particular the differences between the specificity of hNPSF and GnIH at NPFFR2 and an AlphaFold model of NPFFR1. Mutating three residues in NPFFR1 was sufficient to change its specificity of peptide binding to that observed in NPFFR2, corroborating their rationale for their hypothesis. Comparisons are also made with other related neuropeptide receptors, highlighting their similarities but also regions that define the specificity of peptide binding. The authors used the inactive ligand-free state to show that NPFFR2 activation closely follows the canonical activation pathway for Class A GPCRs. In addition, mutagenesis of the residues lining the orthosteric binding site decrease the ability of the receptor to activate G protein.

The structures are of reasonable quality and are sufficient to draw the conclusions described in the manuscript. The manuscript is written clearly and is well supported by the figures. There are only a few minor comments that need addressing.

1. In the panels for Fig 1A, 4B and 4D, please insert the errors for the EC50 values.
2. In Fig 3, it would be nice to have a direct comparison between the positions of residues that make contact to the ligands in all the related receptors. It would make the discussion in the text much easier to follow. An example of such a figure is Fig. 2c in <https://doi.org/10.1038/s41467-024-51793-w>, although I am sure the authors can probably think of something even better.
3. In Fig 2B, it is fine to show the delta pEC50, but all the actual values need to be available as a table in the Appendix, and should include Emax, EC50, cell surface expression, errors and P values for the comparisons. There also needs to be a mention in the Figure legend to this table and also to the Appendix Fig. 2B where the raw data are plotted.
4. In the figure legends of the structures (e.g. Fig 1B, Fig 3A, Fig 5A) the cartoons are named in a list, followed by their respective colours. This is fine when there are only two cartoons, but when there are eight, it is not clear. Please reformulate e.g. ...cartoon models: NPFFR2 (green); hNPSF (yellow); etc
5. All resolutions and RMSDs should be cited to only one decimal place. The error of the structures is probably around 0.3 Å (or greater), so the second decimal place is superfluous.
6. Page 6 line 2: the explanation that the size of the G protein-receptor interface is related to the narrowness of the cleft is over simplistic. For example, there are usually interactions mediated by the G protein outside the alpha5 helix which can make contacts to ECL2 and ECL3, and some interactions may occur to the beta subunit. See <https://doi.org/10.1016/j.mce.2019.02.006>
7. Page 8, para 2. Please add the RMSDs between NPFFR2 and the related receptors.
8. Page 8, para 3, line 6; 'admide' should be 'amide'
9. Page 11, para 1, first sentence: add 'in the active state' at the end of the sentence to improve clarity.
10. Page 11, para 1, line 5, should be 'Greek beta'-strands.
11. Page 20 final line, the last sentence is a duplicate of the previous sentence.

Referee #2:

The authors describe the first cryo-EM structures of the neuropeptide FF receptor 2 in the active Gi-coupled and ligand-free states. Using these models, they seek to understand the structural basis for peptide selectivity at NPFFR2 and related GPCRs, as well as a mechanism of activation for the NPFFR2.

Given my expertise, I have focused predominantly on the cryo-EM and structural biology data and interpretation for this manuscript review. Overall the cryo-EM data is of good quality and the conclusions drawn are convincing. More specifically:

- Modes of binding of different RF-amide peptides to the NPFFR2 and related GPCRs.
- mechanism for peptide selectivity between NPFFR1 and NPFFR2 outlined and supported by functional mutagenesis.
- Structural differences between active and inactive states of NPFFR2.

However, there are many sections of the paper that are confusing to read, in particular for scientists outside the NPFFR field. See some examples of this listed below. The paper also has a few remaining spelling and grammar issues, some of which are

outlined below.

Summary:

1. Does this manuscript report a single key finding? YES

Structural basis for ligand recognition and specificity in NPFFR2.

2. Is the reported work of significance (YES), or does it describe a confirmatory finding or one that has already been documented using other methods or in other organisms etc (NO)? YES

3. Is it of general interest to the molecular biology community? YES

The delineation of the mechanism of RF-amide peptide selectivity between NPFFR1 and 2 seems particularly useful for structure-based drug design in order to specifically target one of these receptors. The cryo-EM structure of the ligand-free receptor is also of particular technical interest, given it is still difficult to solve small membrane proteins using cryo-EM.

4. Is the single major finding robustly documented using independent lines of experimental evidence (YES), or is it really just a preliminary report requiring significant further data to become convincing, and thus more suited to a longer format article (NO)? YES

Overall, this was an interesting paper to read and I congratulate the authors on their apo/ligand-free structure. I invite the authors to revise the manuscript to address specific concerns listed below.

In regard to assessing the quality of cryo-EM data: Overall, many residues and their interactions, e.g. ligand-receptor, that the authors discuss throughout the paper are mostly supported by the consensus and associated half maps. As such the structural conclusions made based on these models appear sound. However, the following points need clarification and should be addressed in a revised manuscript version.

Points to address in regard to cryo-EM data:

- Major data quality concern: How were the main maps provided ('NPFFR2_active.mrc' / 'NPFFR2_apo.mrc') generated? The receptor portion of the 'NPFFR2_active.mrc' map appears to be very similar to the 'active_local_receptor_sharp.mrc', though the G protein is not. Where multiple maps 'fused' in to a composite map? A similar discrepancy is apparent for the apo maps. It is not the same as any of the global or locally refined apo maps provided. This was not clarified in the methods or in the supplementary data, therefore this needs to be further explained in the manuscript. Furthermore, the receptor signal in both the 'apo_local_receptor_sharp.mrc' and 'apo_local_receptor_unsharpened.mrc' maps is quite weak relative to the other parts of the maps, suggesting the receptor was not the focus of these refinements. This is confusing considering the provided 'apo_local_receptor_mask.mrc'. Could the authors clarify how these local refinement maps were generated?
- In general, it is also unclear which maps were used for modelling, just the consensus or the sharpened/locally refined maps as well? All maps used for modelling should be deposited in the PDB/EMDB deposition as additional maps, and the methods should clarify which (region of the) model was built into which map.
- A discussion of the (resolution) limitations of these maps is in general missing, which would provide a more robust assessment of these data. E.g. in figure 5 D the authors show that the Q125 side chain shifts towards the ECD when the hNPSF ligand binds. Whilst this side chain is well resolved in the active map, it is not in the apo map. As such, the modelling of this residue's side chain is ambiguous. I would suggest the authors acknowledge this when discussing the differences in the apo and active maps, given this residue (and others) are extensively discussed. This also applies to Y190, L39, Q308, and I312. K268 of NPFFR2, interacting with Gi, is also not clearly resolved. I would also consider stubbing residues for regions in the apo model that lack higher resolution, i.e. ECL2 and N terminus, and just leaving the backbone.

Other points to address:

- In figure 1A it is unclear which agonist has the lowest potency at each receptor, GnIH or NPVF. There is a similar point of confusion in figure 4B for the same agonists. The values in the legend are inconsistent with the curves. Could the authors clarify this?
- The authors briefly mention an MD study undertaken apparently to identify water-mediated interactions of hNPSF with NPFFR2. While this is interesting, it feels a bit disconnected from the rest of the manuscript and scientific questions asked. I am curious whether the authors considered to run equilibrium (+/- ligand) or unbinding/binding MD simulations to address the limitations of the static cryo-EM data and whether the dynamics differed between the ligand bound/free states.
- The authors claim to have "uncovered the activation mechanism of NPFFR2". Whilst structural comparisons of inactive and active states provides some insight into how a GPCR is activated, it is not the whole picture. Receptor dynamics play an important role in this process, which are not discussed here. Perhaps the author could rephrase to something like 'these structural differences provide insight into the activation mechanism'.
- In the first introduction paragraph it says "Of note, among these five RF-amide peptides, NPFFs and RFRPs share the C-terminal four amino acids, resulting in limited cross-reactivity between NPFFR2 and NPFFR1 toward these peptides." But isn't this similarity in the C terminal portion of these peptides what makes them able to bind these two receptors?
- Results: "In contrast, the ECL2 of NPFFR2 consisting of three β -strands, β 1, β 2, and β 3, displays a distinctive feature that β 2 bends approximately 90 degrees from β 1 and forms a β -sheet with β 3 and the N-terminal β -strand containing residues 32-36 (Fig. 1C)" - In the structure, it appears there are four beta-strands in ECL2, and an additional beta strand that interacts with the N terminus (5 in total in the extracellular region). Beta2 seems unusually long. Would the authors be able to clarify this? A display of the structure in chimeraX using 'dssp' command shows the long beta sheet separated into two.
- In appendix figure S5 I would argue that the N311A mutant doesn't have similar expression levels to WT; it appears to be roughly ~65% of WT.
- In the 'Key interactions of hNPSF determining the ligand-receptor specificity' section there is this sentence 'Notably, NPFF and

RFRPs, which differ only in the N-terminal four residues, selectively activates NPFFR2 and NPFFR1, respectively (Fig. 1A and 4B). But from Fig 4A it seems to me these two classes of peptides differ in more than just the four N terminal residues. E.g. positions -8 and -5 are different across these peptides yet aren't part of the four N terminal residues for some of these peptides.

- In 'Recognition of RF-amide peptide by NPFFR2' section it says 'The two Phe residues, F(-8)SF and F(-6) SF stack together, forming a hydrophobic interaction network with V33Nt' though in the structure model the 33rd residue is a threonine.
- In the 'Key interactions of hNPSF determining the ligand-receptor specificity' section there is this sentence 'However, in NPFFR1, the residues corresponding to the water pocket in NPFFR2 10 (Appendix Fig. S8)'. I believe you are referring to Appendix Fig. S9.

Minor/optional comments.

- The authors state they determined a "ligand-free apo state structure of NPFFR2". I would argue that the structure is not apo as it contains a bRIL fusion and stabilising FABs and Nbs. I suggest calling it a ligand-free structure, or at least offer an explanation at the start of the manuscript and keeping the nomenclature consistent throughout.
- The authors chose many examples/draw comparison to many structures of other receptors throughout the manuscript, but oftentimes they do not clarify the rationale. E.g.: Introduction: "NPFF receptor 2 (NPFFR2, also known as GPR74), NPFF receptor 1 (NPFFR1, also known as GPR147), PrRP receptor (PRLHR), QRFP receptor (QRFPR), and KISS1 receptor (KISS1R), respectively." Results: "Among class A GPCRs bound to peptide ligands, NPYRs exhibit high sequence similarity with NPFFR2,". While this is usually commendable, it makes the manuscript quite difficult to read for non-experts of these receptor types. The authors can consider keeping these descriptions/comparisons or only focus on a handful of reasonable examples to compare their structures to, or think about a figure display or receptor/peptide family tree?
- Consider using both residue numbers and conserved nomenclature for the annotation of G protein residues (as done for receptor residues).

Spelling/grammar:

In the third introduction paragraph the word 'exhibit' needs an 's' on the end and an 'an' after it: "For instance, RF9, which is classified as a partial agonist, often exhibits an antagonist like effect in some experimental models".

In the first sentence of the second paragraph of 'Overall structure of agonist-bound NPFFR2 coupled with Gi', an 'a' is needed between 'to' and 'G protein': "The overall structure of hNPSF-bound NPFFR2 represents the canonical active conformation of class A GPCRs coupled to a G protein".

In 'Conformational changes of NPFFR2 upon activation' section the word 'interact' in this sentence needs an 's' on the end: "Q3087.28 forms a hydrogen bond with Y190ECL2 and I3127.32 interacts with L39Nt which enhances the stability between ECL2 and the TM regions, securing the ligand in place and potentially facilitating ligand activation by stabilizing the receptor's active conformation."

In the discussion the word 'binding' is used twice in this sentence, where the second one should be removed: "In addition, binding of the RF-amide motif binding in the TMD pocket triggers TM3- mediated conformational changes, leading to the overall activation of the receptor."

Some of the in-text citations have extra brackets that aren't necessary, e.g.: "(Li et al., 2016; Xu et al, 2020)".

In the all-atom MD simulations method section there is an apostrophe that should be deleted: "'In the final model of the hNPSF-bound NPFFR2 structure, due to poor EM map quality, residues N-terminal to 24 and C-terminal to 348 were excluded."

We thank the reviewers for their suggestions and comments.

We greatly appreciate their time and efforts in reviewing our manuscript. We believe that our revised manuscript has benefited from their insightful suggestions.

We have addressed every point raised by the reviewers in detailed point-by-point responses (written in blue),

REVIEWER COMMENTS

Referee #1:

Kim et al have determined the cryo-EM structure of an active state of the neuropeptide FF receptor 2 (NPFFR2) bound to the agonist peptide hNDSF and coupled to heterotrimeric G_i , and also the cryo-EM structure of an inactive ligand-free state. The authors use the structure to identify regions that define the specificity of peptide binding, in particular the differences between the specificity of hNPSF and GnIH at NPFFR2 and an AlphaFold model of NPFFR1. Mutating three residues in NPFFR1 was sufficient to change its specificity of peptide binding to that observed in NPFFR2, corroborating their rationale for their hypothesis. Comparisons are also made with other related neuropeptide receptors, highlighting their similarities but also regions that define the specificity of peptide binding. The authors used the inactive ligand-free state to show that NPFFR2 activation closely follows the canonical activation pathway for Class A GPCRs. In addition, mutagenesis of the residues lining the orthosteric binding site decrease the ability of the receptor to activate G protein.

The structures are of reasonable quality and are sufficient to draw the conclusions described in the manuscript. The manuscript is written clearly and is well supported by the figures.

We appreciate the reviewer's comments on our study.

There are only a few minor comments that need addressing.

1. In the panels for Fig 1A, 4B and 4D, please insert the errors for the EC50 values.

We thank the reviewer's comment. We have included the EC50 values with their corresponding errors ($EC_{50} \pm S.E.M.$) into Figures 1A, 4B, and 4D. Additionally, the expanded Table EV1 now includes a comprehensive summary of E_{max} , EC50, and surface expression values, along with their respective errors.

2. In Fig 3, it would be nice to have a direct comparison between the positions of residues that make contact to the ligands in all the related receptors. It would make the discussion in the text much easier to follow. An example of such a figure is Fig. 2c in <https://doi.org/10.1038/s41467-024-51793-w>, although I am sure the authors can probably think of something even better.

We thank the reviewer's suggestion. We have added a new figure, Fig. 3c, highlighting the conserved RF-amide interactions of each ligand with its receptor.

Fig. 3C

RF-amide	R(-2)										F(-1)						amide					
Residue No.	3,33	4,60	4,64	46,52	5,35	5,38	6,39	6,58	6,59	3,33	3,36	5,42	5,46	6,51	6,52	6,55	2,57	2,60	2,61	3,32	7,39	7,43
NPFFR2-hNPSF				F	R		T		D		V	L			W		C		T	Q	H	F
NPFFR1				F	R		T		D		V	L			W		C		T	Q	H	F
QRFP2-QRFP26				F	Q	Y	T		E		V	I	L	F			C		T	Q	Q	F
KISS1R-Kisspeptin10	Q		H			F					V			I		L		F		Q	H	Y
PrRPR-PrRP20		A		F		Y		R		P				L		N			T	Q	H	

Conservation score

31-40	41-50	51-60	61-70	71-80	81-90	91-100
-------	-------	-------	-------	-------	-------	--------

3. In Fig 2B, it is fine to show the delta pEC50, but all the actual values need to be available as a table in the Appendix, and should include Emax, EC50, cell surface expression, errors and P values for the comparisons. There also needs to be a mention in the Figure legend to this table and also to the Appendix Fig. 2B where the raw data are plotted.

We thank the reviewer's comment. As suggested by the reviewer, the actual pEC50 values, as well as Emax, EC50, surface expression data, associated errors, and p-values, have been consolidated in Table EV1. Additionally, the legend for Figure 2B has been revised to direct readers to this table and Appendix Figures S3 and S4, where the raw data are presented.

- Fig. 2 (B) Emax and EC50 values along with their respective errors are summarized in Table EV1.

4. In the figure legends of the structures (e.g. Fig 1B, Fig 3A, Fig 5A) the cartoons are named in a list, followed by their respective colours. This is fine when there are only two cartoons, but when there are eight, it is not clear. Please reformulate e.g. ...cartoon models: NPFFR2 (green); hNPSF (yellow); etc

We thank the reviewer's comment. The figure legends have been updated to clearly associate each cartoon model with its corresponding color, as suggested.

- Fig. 1 (B) The overall cryo-EM electron density map and structural model of the NPFFR2-hNPSF-Gαβγ-scFv16 complex are presented, with the following color scheme. NPFFR2 (green), hNPSF (yellow), Gαi (cyan), Gβ (orange), Gγ (purple), and scFv16 (gray).
- Fig. 3 (A) The unique ECL2 structures and ligand-binding modes are illustrated with cartoon models: NPFFR2 (green), hNPSF (yellow), QRFP26 is (blue-purple), QRFP26 (light brown), Kisspeptin-10 (light cyan), KISS1R (salmon), PrRP20 (pale green), and PrRPR (orange).
- Fig. 5 (A) The cryo-EM density map and structural model of the NPFFR2-BRIL-Fab-Nb complex are displayed, with the following color scheme: ligand-free NPFFR2 (orange), BRIL (cyan), heavy chains of anti-BRIL Fab (Fab^{BRIL}) (yellow), light chains of Fab^{BRIL} (purple), and anti-Fab nanobody (Nb^{Fab}) (gray).

5. All resolutions and RMSDs should be cited to only one decimal place. The error of the structures is probably around 0.3 Å (or greater), so the second decimal place is superfluous.

We thank the reviewer's comment. We have adjusted all reported resolutions and RMSD values to a single decimal place to reflect appropriate precision for the structural data.

6. Page 6 line 2: the explanation that the size of the G protein-receptor interface is related to the narrowness of the cleft is over simplistic. For example, there are usually interactions mediated by the G protein outside

the alpha5 helix which can make contacts to ECL2 and ECL3, and some interactions may occur to the beta subunit. See <https://doi.org/10.1016/j.mce.2019.02.006>

We agree with the reviewer's comment that explaining G-protein selectivity solely through the narrowness of the cleft is overly simplistic. While the ICL3 of NPFFR2 may interact with the G-alpha and -beta subunits, our cryo-EM map could not fully resolve this region for accurate structural analysis. Additionally, in comparison with the active structure of Kiss1R, which was resolved in complex with a chimeric Gq protein (combining elements of Gs, Gi, and Gq), the analysis of Gi and Gq selectivity within the RF-amide receptor family using this structure was further constrained. Therefore, the corresponding sentence was removed. Instead, we focused our analysis on the interactions between the alpha5 helix of the G-alpha subunit and the receptor. These limitations have been clearly addressed and incorporated into the main text.

Page 5-6; The binding interface between NPFFR2 and $G\alpha_i$ measures 774 \AA^2 , in contrast to the more extensive $1,455 \text{ \AA}^2$ interface observed in the KISS1R and $G\alpha_{qNi}$ complex. In KISS1R, the cytoplasmic regions of TMD and intracellular loop 2 (ICL2) are involved in the interactions with the $\alpha 5$ helix, the αN and $\alpha 5$ regions, and the $\alpha 4/\beta 6$ regions of the $G\alpha_q$ subunit. By comparison, the interaction between NPFFR2 and $G\alpha_i$ is primarily mediated through the $\alpha 5$ helix of the $G\alpha_i$ subunit and the TMD of the receptor.

7. Page 8, para 2. Please add the RMSDs between NPFFR2 and the related receptors.

We thank the reviewer's comment. The RMSD values between NPFFR2 and related receptors have been included in the main text.

Page 8; Structural alignment of the TMDs of NPFFR2 with those of these RF-amide receptors revealed RMSD values of 1.1 \AA for QRFPR, 1.3 \AA for KISS1R, and 1.0 \AA for PrRPR, indicating high structural similarity in their TMDs.

8. Page 8, para 3, line 6; 'admide' should be 'amide'

Thank you for pointing out the error. The mistake has been corrected in the manuscript.

*Page 8; (superscript RF refers to the residue from RF-**amide** peptides in general)*

9. Page 11, para 1, first sentence: add 'in the active state' at the end of the sentence to improve clarity.

We thank the reviewer's comment. The sentence was revised as suggested.

*Page 11; such as 7 \AA outward movement of the cytoplasmic segment of TM6 **in the active state***

10. Page 11, para 1, line 5, should be 'Greek beta'-strands.

Thank you for pointing this out. We revised the sentence as suggested.

*Page 11; the **β -strands** of ECL2*

11. Page 20 final line, the last sentence is a duplicate of the previous sentence.

Thank you for pointing out the error. The mistake has been corrected in the manuscript.

Referee #2:

The authors describe the first cryo-EM structures of the neuropeptide FF receptor 2 in the active Gi-coupled and ligand-free states. Using these models, they seek to understand the structural basis for peptide selectivity at NPFFR2 and related GPCRs, as well as a mechanism of activation for the NPFFR2.

Given my expertise, I have focused predominantly on the cryo-EM and structural biology data and interpretation for this manuscript review. Overall the cryo-EM data is of good quality and the conclusions drawn are convincing. More specifically:

- Modes of binding of different RF-amide peptides to the NPFFR2 and related GPCRs.
- mechanism for peptide selectivity between NPFFR1 and NPFFR2 outlined and supported by functional mutagenesis.
- Structural differences between active and inactive states of NPFFR2.

However, there are many sections of the paper that are confusing to read, in particular for scientists outside the NPFFR field. See some examples of this listed below. The paper also has a few remaining spelling and grammar issues, some of which are outlined below.

Summary:

1. Does this manuscript report a single key finding? YES

Structural basis for ligand recognition and specificity in NPFFR2.

2. Is the reported work of significance (YES), or does it describe a confirmatory finding or one that has already been documented using other methods or in other organisms etc (NO)? YES

3. Is it of general interest to the molecular biology community? YES

The delineation of the mechanism of RF-amide peptide selectivity between NPFFR1 and 2 seems particularly useful for structure-based drug design in order to specifically target one of these receptors. The cryo-EM structure of the ligand-free receptor is also of particular technical interest, given it is still difficult to solve small membrane proteins using cryo-EM.

4. Is the single major finding robustly documented using independent lines of experimental evidence (YES), or is it really just a preliminary report requiring significant further data to become convincing, and thus more suited to a longerformat article (NO)? YES

Overall, this was an interesting paper to read and I congratulate the authors on their apo/ligand-free structure. I invite the authors to revise the manuscript to address specific concerns listed below.

In regard to assessing the quality of cryo-EM data: Overall, many residues and their interactions, e.g. ligand-receptor, that the authors discuss throughout the paper are mostly supported by the consensus and associated half maps. As such the structural conclusions made based on these models appear sound. However, the following points need clarification and should be addressed in a revised manuscript version.

Points to address in regard to cryo-EM data:

1. Major data quality concern: How were the main maps provided ('NPFFR2_active.mrc' / 'NPFFR2_apo.mrc') generated? The receptor portion of the 'NPFFR2_active.mrc' map appears to be very similar to the 'active_local_receptor_sharp.mrc', though the G protein is not. Were multiple maps 'fused' in to a composite map? A similar discrepancy is apparent for the apo maps. It is not the same as any of the global or locally refined apo maps provided. This was not clarified in the methods or in the supplementary data, therefore this needs to be further explained in the manuscript. Furthermore, the receptor signal in both the 'apo_local_receptor_sharp.mrc' and 'apo_local_receptor_unsharpened.mrc' maps is quite weak relative to the other parts of the maps, suggesting the receptor was not the focus of these refinements. This is confusing considering the provided 'apo_local_receptor_mask.mrc'. Could the authors clarify how these local refinement maps were generated?

We thank the reviewer's comment.

(1) Map generation information.

The following provides detailed information on the map generation process:

- Initially, the global maps ('active_global_unsharpened.mrc' and 'apo_global_unsharpened.mrc') were generated through non-uniform (NU) refinement following 3D classification.
- Map sharpening was subsequently performed to create 'active_global_sharp.mrc' and 'apo_global_sharp.mrc'. These two maps were deposited to the EMDB.
- The 'active_global_sharp.mrc' map was further locally refined using 'active_local_receptor_mask.mrc' and 'active_local_Gprotein_mask.mrc', to generate 'active_local_receptor_sharp.mrc' and 'active_local_Gprotein_sharp.mrc', respectively. These two maps were combined using vop maximum to generate the composite map (NPFFR2_active.mrc), which we deposited to the EMDB as an additional map.
- Similarly, the 'apo_global_sharp.mrc' map was locally refined to generate the 'apo_local_receptor_sharp.mrc' and 'apo_local_BRIL-Fab_sharp.mrc' maps. Then, these two maps were combined using vop maximum to generate the composite map (NPFFR2_apo.mrc), which we deposited to the EMDB as an additional map.

As suggested by the reviewer, we have included this detailed data processing method in the Methods section for clarity.

Page 17-18; Non-uniform (NU) refinement was performed in cryoSPARC using default settings, yielding the final maps with global nominal resolutions of 3.2 Å for the hNPSF–NPFFR2–Gi complex and 2.9 Å for the NPFFR2–BRIL– Fab^{BRIL}–Nb^{Fab} complex, as determined by the gold-standard Fourier Shell Correlation (GSFSC) at the 0.143 criterion. To facilitate model building, local refinements were performed separately for NPFFR2, the Gi–scFv16 complex, and the BRIL–Fab^{BRIL}–Nb^{Fab} complex G-protein, and the BRIL–Fab–Nb complex using individual masks. These refinements employed pose/shift Gaussian priors during alignment, with a rotation standard deviation of 5 degrees and a shift standard deviation of 2 Å.

(2) Addressing the concern of weak receptor signal

Regarding the concern about the weak receptor signal in both the 'apo_local_receptor_sharp.mrc' and 'apo_local_receptor_unsharpened.mrc' maps, we checked the map files provided to the reviewers. Upon re-examination, we realized that a simple labeling error occurred: the local maps of the receptor and BRIL-Fab were accidentally swapped, resulting in incorrect labeling. Consequently, the "BRIL-Fab" map was mistakenly provided as the receptor map (see figure below). This explains the reviewer's observation that "the receptor signal in both the 'apo_local_receptor_sharp.mrc' and 'apo_local_receptor_unsharpened.mrc' maps is quite weak relative the other parts of the maps, suggesting the receptor was not the focus of these refinements." (see figure below). We sincerely apologize for this mistake and any confusion it may have caused. Now, we have corrected the filenames and ensured the accuracy of the provided files. The corrected receptor-focused map files are: "apo_local_recptor.mrc", "apo_local_receptor_sharp.mrc", "apo_local_receptor_half_A.mrc", and "apo_local_receptor_halfB.mrc". Additionally, we have included the previously missing receptor-focused mask file, "apo_receptor_mask.mrc". All these renamed files have been uploaded for review.

(<https://www.dropbox.com/scl/fo/u9ufg7slkw5sargkox096/ADGXpPaeX62VmPBABi-3uxw?rlkey=rfkqffrxtyv6vyq72uh5j0fan&e=1&st=fdq734ro&dl=0>)

2. In general, it is also unclear which maps were used for modelling, just the consensus or the sharpened/locally refined maps as well? All maps used for modelling should be deposited in the PDB/EMDB deposition as additional maps, and the methods should clarify which (region of the) model was built into which map.

Model building and refinement were performed using the sharpened composite maps ('NPFFR2_active.mrc', 'NPFFR2_apo.mrc'), while validation was carried out using the global maps ('active_global_sharp.mrc', 'apo_global_sharp.mrc'). The global maps were submitted as the primary maps to the EMDB, with the composite maps included as additional maps. These details have been clarified and elaborated in the Methods section.

Page 18; Model buildings and refinement were conducted using the locally refined sharp maps.

Page 20; The sharpened composite maps, generated by combining the locally refined maps, have been deposited in the EMDB as additional maps.

3. A discussion of the (resolution) limitations of these maps is in general is missing, which would provide a more robust assessment of these data. E.g. in figure 5 D the authors show that the Q125 side chain shifts towards the ECD when the hNPSF ligand binds. Whilst this side chain is well resolved in the active map, it is not in the apo map. As such, the modelling of this residue's side chain is ambiguous. I would suggest the authors acknowledge this when discussing the differences in the apo and active maps, given this residue (and others) are extensively discussed. This also applies to Y190, L39, Q308, and I312. K268 of NPFFR2,

interacting with G_i , is also not clearly resolved. I would also consider stubbing residues for regions in the apo model that lack higher resolution, i.e. ECL2 and N terminus, and just leaving the backbone.

We thank the reviewer's suggestion.

- We agree that side chains should not be modeled when their corresponding density is not clear. Consequently, ambiguous side chains in the ECL and ICL regions of both the apo and active structures have been removed in the coordinate files (PDPs are updated) and Figure 5.
- Regarding the Q125 side chain: Although it is not fully resolved in the apo map, we could infer that Q125 undergoes an upward rotamer shift upon peptide binding to avoid steric clash with $F(-1)^{SF}$ and to facilitate interaction with the peptide's C-terminal amide group. This inference is based on observable side chain density up to the C_γ atom of Q125 (see figure below, Appendix Figure S12).

The manuscript has been revised to reflect the limited density for Q125 and other residues (e.g., L39, Y190, Q308, I312) that undergo conformational changes during activation.

Page 11; *Several residues involved in hNPSF binding or forming the internal interaction network in the active state, such as L39^{Nt}, Y190^{ECL2}, Q308^{7,28}, and I312^{7,32} exhibit poorly defined side chain density in the ligand-free state map, suggesting high flexibility of these residues in the absence of agonist.*

At the bottom of the ligand binding pocket, the side chain of Q125^{3,32} is well-resolved in the active state, where it forms a polar interaction with the C-terminal amide of hNPSF. In the ligand-free state, this side chain density is less-resolved (figure). Upon hNPSF binding, $F(-1)^{SF}$ occupies the position where Q125^{3,32} is located in the ligand-free state, leading to an upward rotamer shift of Q125^{3,32} to avoid steric clash with $F(-1)^{SF}$.

- Regarding K268 of NPPFR2: While the side chain of K268 is not fully resolved, we observe density for its C_γ and $C\delta$ carbons, which form van der Waals interactions with $F354^{H5,26}$ of $G\alpha_i$ (see figure below). We clarify this interaction in the revised manuscript.

Page 6; *While $F354^{H5,26}$ of $G\alpha_i$ interacts with NPPFR2 by forming van der Waals contacts with the C_γ and $C\delta$ carbons of $K268^{6,29}$, the corresponding residue $V361^{H5,26}$ in $G\alpha_{qNi}$ does not establish any contact with KISS1R.*

Other points to address:

4. In figure 1A it is unclear which agonist has the lowest potency at each receptor, GnIH or NPVF. There is a similar point of confusion in figure 4B for the same agonists. The values in the legend are inconsistent with the curves. Could the authors clarify this?

We thank the reviewer for pointing out this errors and apologize for the mistake. The EC₅₀ values for the ligands GnIH and NPVF were accidentally swapped in Figures 1A, 4B, and 4D. Now, we have corrected the EC₅₀ values as follows:

- Figure 1A: GnIH: 680 nM → 140 nM, NPVF: 140 nM → 680 nM
- Figure 4B: GnIH: 2.0 nM → 3.0 nM, NPVF: 3.0 nM → 2.0 nM
- Figure 4D: GnIH: 2.0 nM → 3.0 nM

The corrected EC₅₀ values have also been included in Table EV1 for clarity.

5. The authors briefly mention an MD study undertaken apparently to identify water-mediated interactions of hNPSF with NPFFR2. While this is interesting, it feels a bit disconnected from the rest of the manuscript and scientific questions asked. I am curious whether the authors considered to run equilibrium (+/- ligand) or unbinding/binding MD simulations to address the limitations of the static cryo-EM data and whether the dynamics differed between the ligand bound/free states.

We conducted MD simulations for both the ligand-bound and ligand-free states. In the ligand-free state, increased movement of the N-terminal β-strand, ECL2 and ECL3 regions was observed compared to the ligand-bound-state. These dynamics reflect the inherent flexibility of the receptor in the ligand-free state and likely contribute to the lower resolution observed in its structure. This observation aligns with previous findings for Class A GPCRs, which are known to exhibit high structural dynamics in the absence of a ligand. Since the MD simulations did not reveal significant novel insights beyond these established dynamics and were somewhat disconnected from the main focus of our manuscript (as noted by the reviewer), we chose not to include an extensive discussion of the structural dynamics for the two states in the manuscript. Instead, we focused on examining the importance of polar interactions near Q(-5) of hNPSF. Through these simulations, we identified water-mediated interactions involving Q(-5), which we believe provide valuable insights into ligand selectivity. This finding directly supports the main argument of our study.

6. The authors claim to have "uncovered the activation mechanism of NPFFR2". Whilst structural comparisons of inactive and active states provides some insight into how a GPCR is activated, it is not the whole picture. Receptor dynamics play an important role in this process, which are not discussed here. Perhaps the author could rephrase to something like 'these structural differences provide insight into the activation mechanism'.

We agree with the reviewer's comment. The sentence was rephrased to reflect the reviewer's suggestion.

Page 12; Furthermore, these structural differences provide insight into the activation mechanism of NPFFR2 upon ligand binding by comparing its ligand-free and active structures.

7. In the first introduction paragraph it says "Of note, among these five RF-amide peptides, NPFFs and RFRPs share the C-terminal four amino acids, resulting in limited cross-reactivity between NPFFR2 and NPFFR1 toward these peptides." But isn't this similarity in the C terminal portion of these peptides what makes them able to bind these two receptors?

We thank the reviewer's comment. We used the term "limited cross-reactivity" to represent "sharing binding property" but not fully interconvertible. However, this term could be misinterpreted as indicating minimal or negligible cross-reactivity. Therefore, we revise the sentence for better precision as below.

Page 3; Notably, among these five RF-amide peptides, NPFFs and RFRPs share the C-terminal four amino acids, resulting in cross-reactivity between NPFFR2 and NPFFR1, albeit with differing binding affinities.

8. Results: "In contrast, the ECL2 of NPFFR2 consisting of three β -strands, β 1, β 2, and β 3, displays a distinctive feature that β 2 bends approximately 90 degrees from β 1 and forms a β -sheet with β 3 and the N-terminal β -strand containing residues 32-36 (Fig. 1C)" - In the structure, it appears there are four beta-strands in ECL2, and an additional beta strand that interacts with the N terminus (5 in total in the extracellular region). Beta2 seems unusually long. Would the authors be able to clarify this? A display of the structure in chimeraX using 'dssp' command shows the long beta sheet separated into two.

We agree with the reviewer's comment. Residues 202 and 203 of ECL2 were initially identified as part of a beta structure in the Ramachandran plot. However, as these residues do not interact with the neighboring beta strand, the long beta strand (198-208) was reclassified into two separate beta strands (198-201, 204-208). Thus, structural description of the ECL2 has been revised as to have four-beta strands, and the corresponding structural model has been updated accordingly in Figure 1B.

Page 5; By contrast, the ECL2 of NPFFR2 consists of four β -strands, β 2 - β 5, and an additional β 1 strand is formed by residues 32-36 in the N-terminal tail. Notably, β 3 bends approximately 90 degrees from β 2 and forms a β -sheet with β 1 and β 4, while β 2 and β 5 forms another distinct β -sheet (Fig. 1C).

9. In appendix figure S5 I would argue that the N311A mutant doesn't have similar expression levels to WT; it appears to be roughly ~65% of WT.

We thank the reviewer's comment. To address this concern, we have included cAMP response data for the WT receptor, where the expression level was reduced to 50% (similar level to N311A expression) by transfecting half the amount of DNA. The results indicate that the EC50 value for the WT at 50% expression is comparable to that at 100% expression (see figure below). This finding confirms that the reduced Gi signaling observed in the N311A mutant is not caused by its relatively lower expression level. This additional data (WT 50% DNA transfected) has been incorporated into Fig. 2B for comparison.

ELISA-based surface expression assay **GloSensor™ cAMP assays**

10. In the 'Key interactions of hNPSF determining the ligand-receptor specificity' section there is this sentence 'Notably, NPFF and RFRPs, which differ only in the N-terminal four residues, selectively activates NPFFR2 and NPFFR1, respectively (Fig. 1A and 4B).' But from Fig 4A it seems to me these two classes of peptides differ in more than just the four N terminal residues. E.g. positions -8 and -5 are different across these peptides yet aren't part of the four N terminal residues for some of these peptides.

Thank you for pointing out the error. The incorrect sentence has been revised to state that the residues at positions -5 to N-terminus are different.

Page 9; Notably, NPFFs and RFRPs, which differ in the residues at position -5 to N-terminus, ---

11. In 'Recognition of RF-amide peptide by NPFFR2' section it says 'The two Phe residues, F(-8)SF and F(-6) SF stack together, forming a hydrophobic interaction network with V33Nt' though in the structure model the 33rd residue is a threonine.

Thank you for pointing out the error. The residue numbering in the main text was incorrect and has been corrected from V33^{Nt} to V35^{Nt}.

Page 7; forming a hydrophobic interaction network with V35^{Nt}, L39^{Nt}, Y190^{ECL2}, ---

12. In the 'Key interactions of hNPSF determining the ligand-receptor specificity' section there is this sentence 'However, in NPFFR1, the residues corresponding to the water pocket in NPFFR2 10 (Appendix Fig. S8)'. I believe you are referring to Appendix Fig. S9.

Thank you for pointing out the error. The error has been corrected, with the reference updated from Appendix Fig. S8 to Appendix Fig. S9.

Page 9; the residues corresponding to the water pocket in NPFFR2 (Appendix Fig. S9)

Minor/optional comments.

13. The authors state they determined a "ligand-free apo state structure of NPFFR2". I would argue that the structure is not apo as it contains a bRIL fusion and stabilising FABs and Nbs. I suggest calling it a ligand-free structure, or at least offer an explanation at the start of the manuscript and keeping the nomenclature consistent throughout.

We agree with the reviewer's comment. The term "apo structure" has been replaced with "ligand-free structure" throughout the manuscript to maintain consistency.

14. The authors chose many examples/draw comparison to many structures of other receptors throughout the manuscript, but oftentimes they do not clarify the rationale. E.g.: Introduction: "NPFF receptor 2 (NPFFR2, also known as GPR74), NPFF receptor 1 (NPFFR1, also known as GPR147), PrRP receptor (PRLHR), QRFP receptor (QRFPR), and KISS1 receptor (KISS1R), respectively." Results: "Among class A GPCRs bound to peptide ligands, NPYRs exhibit high sequence similarity with NPFFR2,". While this is usually commendable, it makes the manuscript quite difficult to read for non-experts of these receptor types. The authors can consider keeping these descriptions/comparisons or only focus on a handful of reasonable examples to compare their structures to, or think about a figure display or receptor/peptide family tree?

We thank the reviewer's comment. The RF-amide receptors (excluding Kiss1R) and the NPYR family share approximately 50% sequence similarity and are closely related. Notably, NPFFR2 exhibited the

smallest RMSD when aligned with NPY2R structures, indicating a high degree of structural similarity. To address the concern, the main text has been revised to clarify the rationale behind these comparisons, and a phylogenetic tree of related 78 class A GPCRs has been included as Appendix Figure S1.

Page 3; Among the five RF-amide receptors, KISS1R shows notable phylogenetic divergence, with relatively low sequence similarity to the other four RF-amide receptors (~ 33 %) (Appendix Fig. S1). Interestingly, neuropeptide Y (NPY) receptors, although not part of the RF-amide receptor family, are more closely related to the four RF-amide receptors than KISS1R is, sharing approximately 50% sequence similarity.

Page 5; Interestingly, the hNPSF-NPFFR2-G_i complex exhibits high structural similarity to the NPY-NPY2R-G_i complex (Kang et al, 2023), with an RMSD of 0.9 Å for their transmembrane domains (TMDs) (Fig. EV2A) (Fig. EV2A).

15. Consider using both residue numbers and conserved nomenclature for the annotation of G protein residues (as done for receptor residues).

We thank the reviewer's suggestion. As suggested, we used conserved nomenclature for the annotation of G-protein residues, such as "F354^{H5.26} of G α_i " in page 6.

Spelling/grammar:

16. In the third introduction paragraph the word 'exhibit' needs an 's' on the end and an 'an' after it: "For instance, RF9, which is classified as a partial agonist, often exhibits an antagonist like effect in some experimental models".

Thank you for pointing out the errors. We corrected the grammar errors.

*Page 3; RF9, which is classified as a partial agonist, often exhibits **an** antagonist-like effect --*

17. In the first sentence of the second paragraph of 'Overall structure of agonist-bound NPFFR2 coupled with G_i', an 'a' is needed between 'to' and 'G protein': "The overall structure of hNPSF-bound NPFFR2 represents the canonical active conformation of class A GPCRs coupled to a G protein".

Thank you for pointing this out. The mistake has been corrected.

*Page 5; The overall structure of hNPSF-bound NPFFR2 represents the canonical active conformation of class A GPCRs coupled to **a** G protein.*

18. In 'Conformational changes of NPFFR2 upon activation' section the word 'interact' in this sentence needs an 's' on the end: "Q3087.28 forms a hydrogen bond with Y190^{ECL2} and I3127.32 interacts with L39^{Nt} which enhances the stability between ECL2 and the TM regions, securing the ligand in place and potentially facilitating ligand activation by stabilizing the receptor's active conformation."

Thank you for pointing this out. The mistake has been corrected.

Page 11; Q3087.28 forms a hydrogen bond with Y190^{ECL2} and I3127.32 interacts with L39^{Nt} which --

19. In the discussion the word 'binding' is used twice in this sentence, where the second one should be removed: "In addition, binding of the RF-amide motif binding in the TMD pocket triggers TM3- mediated

conformational changes, leading to the overall activation of the receptor."

Some of the in-text citations have extra brackets that aren't necessary, e.g.: "(Li et al., 2016; (Xu et al, 2020)".

Thank you for pointing out the errors. The mistakes have been corrected in the manuscript (page 12).

20. In the all-atom MD simulations method section there is an apostrophe that should be deleted: " "In the final model of the hNPSF-bound NPFFR2 structure, due to poor EM map quality, residues N-terminal to 24 and C-terminal to 348 were excluded."

Thank you for pointing this out. The unnecessary apostrophe has been removed (page 19).

Dear Prof. Choi,

Thank you for the submission of your revised manuscript. We have now received the enclosed report from referee 2 who was asked to assess it. This referee still has 2 more, minor suggestions that I would like you to incorporate before we can proceed with the official acceptance of your manuscript. Please co-submit a point-by-point response to all final comments/requests.

A few editorial requests will also need to be addressed:

- Please remove the author credits from the ms file. All credits need to be entered during online ms submission.
- One Table EV1 is uploaded; the legend should be removed from the ms file and instead needs to be provided in the Excel file of the table. The correct title/nomenclature should be Table EV1 (instead of Expanded View Table 1).
- APPENDIX FILE: the table of content needs to have each Appendix item listed on the title page with the corresponding page number.
- The source data (SD) for the main figures need to be uploaded as one zip folder per one figure. The SD for EV and Appendix figures should be zipped into one folder.
- The manuscript sections should be in the following order: Title page - Abstract & Keywords - Introduction - Results - Discussion - Methods - Data Availability - Acknowledgments - Disclosure Statement & Competing Interests - References - Figure Legends - (Main Tables with legends if applicable) - Expanded View Figure Legends.
- The nomenclature of the EV figure legends in the ms file is not correct, it should be Figure EV1, etc. instead of Expanded View Figure 1, etc.
- Please provide the specific URLs for EMD-6144, EMD-61446, 9JFY, 9JG0 datasets in the data availability section.
- Please provide the exact p values in the legend of figure 2B (as reasonable).

I would like to suggest some minor changes to the abstract. Can you please address my comments in brackets:

Neuropeptide FF Receptor 2 (NPFFR2), a G protein-coupled receptor, plays a role in pain modulation and diet-induced thermogenesis. While NPFFR2 is strongly activated by neuropeptides FF (NPFFs), it shows low activity [OK?] in response to RF-amide-related peptides (RFRPs), despite the peptides belonging to a shared family [OK?]. In contrast, NPFFR1, which shares high sequence similarity with NPFFR2, is activated by RFRPs and regulates reproductive hormone balance. The molecular basis for these receptor-specific interactions with their RF-amide peptides remains unclear. Here, we present cryo-electron microscopy structures of NPFFR2 in its active state bound to the agonist RF-amide peptide [OK?] hNPSF, and in its ligand-free state. Structural analysis reveals that the C-terminal RF-amide moiety engages conserved residues in the transmembrane domain, while the N-terminal segment interacts in a receptor subtype-specific manner. Key selectivity-determining residues in NPFFR2 are also identified. A homology model of NPFFR1 bound to RFRP, supported by mutagenesis studies, further validates this selectivity mechanism. Additionally, structural comparison between the inactive and active states of NPFFR2 demonstrates a TM3-mediated activation mechanism. These findings provide insights into RF-amide peptide recognition by NPFF receptors.

EMBO press papers are accompanied online by A) a short (1-2 sentences) summary of the findings and their significance, B) 2-3 bullet points highlighting key results and C) a synopsis image that is exactly 550 pixels wide and 200-600 pixels high (the height is variable). The image you sent is good. Please send us the short summary and bullet points with the final manuscript.

Referee #2:

The authors have addressed many of the original comments accurately, however, I still have two (minor) concerns:

Regarding map generating process: The added explanation in the method section is good, though it still lacks a description of the generation of the composite maps. At the moment, it reads as if the maps generated in the local refinement jobs themselves were used, rather than a composite map generated from two locally refined maps.

This is also true of the workflow graphics in Figure EV1 and Figure EV4; they both lack a mention of any composite maps. These should be shown here for clarity, as well as local resolution estimates of these maps. In addition: The sentence added on page 18 reads as if the authors used the locally refined maps for modelling, rather than the composite map as mentioned in the comment in the "Point by point" response file. Please make sure this is clear in the manuscript. The authors mentioned the composite maps on page 20, but still why they were made and for what purpose has not been mentioned yet. If these composite maps were used for modelling further detailed explanation of this process (what software was used) and reasons for it are required.

Regarding stubbing/removing of residues for deposition: It seems highly unusual to keep only some of the side chain atoms and not all/none. E.g. for the case of K268 and Q125 I have not seen this for any PDB files before, and the role of this residue and region seems rather over-interpreted, given the resolution. I would recommend to the authors to stub/remove the side chain and adjust the manuscript text accordingly. E.g. speculations/hypotheses for lower resolution regions are fine, however, making sure that the text clearly reflects the resolution limitations, and also making sure that if the remainder of the atoms were to be modelled, that these would make sense in terms of geometry/chemical restraints or clashes. If the side chains are to be stubbed, then the authors should make sure that the manuscript refers to additional data to support their hypothesis (e.g. functional readouts, MD simulations, etc).

Dear Editor,

Thank you for the comments.

Here is a point-by-point response to all final comments/requests (written in blue).

Editorial requests:

1) Please remove the author credits from the ms file. All credits need to be entered during online ms submission.

We have removed the author credits from the manuscript file as requested.

2) One Table EV1 is uploaded; the legend should be removed from the ms file and instead needs to be provided in the Excel file of the table. The correct title/nomenclature should be Table EV1 (instead of Expanded View Table 1).

The legend for Table EV1 has been removed from the manuscript file and included in the Excel file of Table EV1.

The nomenclature has been updated to "Table EV1".

3) APPENDIX FILE: the table of content needs to have each Appendix item listed on the title page with the corresponding page number.

The table of contents in Appendix file has been updated to list each Appendix item on the title page with the corresponding page number.

4) The source data (SD) for the main figures need to be uploaded as one zip folder per one figure. The SD for EV and Appendix figures should be zipped into one folder.

The source data for the main figures have been uploaded as one zip folder per figure (such as SourceDataForFigure1.zip). The source data for EV and Appendix figures have been zipped into one folder (SourceDataFor EV and Appendix.zip).

5) The manuscript sections should be in the following order: Title page - Abstract & Keywords - Introduction - Results - Discussion - Methods - Data Availability - Acknowledgments - Disclosure Statement & Competing Interests - References - Figure Legends - (Main Tables with legends if applicable) - Expanded View Figure Legends.

We have revised the manuscript to follow the correct section order

6) The nomenclature of the EV figure legends in the ms file is not correct, it should be Figure EV1, etc. instead of Expanded View Figure 1, etc.

The nomenclature of EV figure legends has been corrected to "Figure EV1," "Figure EV2," etc.

7) Please provide the specific URLs for EMD-6144, EMD-61446, 9JFY, 9JG0 datasets in the data availability section.

The specific URLs for EMD-61444, EMD-61446, PDB 9JFY, and PDB 9JG0 have been added to the data availability section (page 21).

8) Please provide the exact p values in the legend of figure 2B (as reasonable).

We have updated the legend of Figure 2B to indicate that the exact p-values are provided in Table EV1.

*Page 26; ns (not significant, $p > 0.05$); * $p < 0.05$; ** $p < 0.01$; *** $p < 0.001$. The exact p-values are provided in Table EV1.*

9) I would like to suggest some minor changes to the abstract. Can you please address my comments in brackets: Neuropeptide FF Receptor 2 (NPFFR2), a G protein-coupled receptor, plays a role in pain modulation and diet-induced thermogenesis. While NPFFR2 is strongly activated by neuropeptides FF (NPFFs), it shows low activity [OK] in response to RF-amide-related peptides (RFRPs), despite the peptides belonging to a shared family [OK]. In contrast, NPFFR1, which shares high sequence similarity with NPFFR2, is activated by RFRPs and regulates reproductive hormone balance. The molecular basis for these receptor-specific interactions with their RF-amide peptides remains unclear. Here, we present cryo-electron microscopy structures of NPFFR2 in its active state bound to the agonist RF-amide peptide [OK] hNPSE, and in its ligand-free state. Structural analysis reveals that the C-terminal RF-amide moiety engages conserved residues in the transmembrane domain, while the N-terminal segment interacts in a receptor subtype-specific manner. Key selectivity-determining residues in NPFFR2 are also identified. A homology model of NPFFR1 bound to RFRP, supported by mutagenesis studies, further validates this selectivity mechanism. Additionally, structural comparison between the inactive and active states of NPFFR2 demonstrates a TM3-mediated activation mechanism. These findings provide insights into RF-amide peptide recognition by NPFF receptors.

We have accepted your suggestions and reflected them in the abstract.

10) EMBO press papers are accompanied online by A) a short (1-2 sentences) summary of the findings and their significance, B) 2-3 bullet points highlighting key results and C) a synopsis image that is exactly 550 pixels wide and 200-600 pixels high (the height is variable). The image you sent is good. Please send us the short summary and bullet points with the final manuscript.

We will submit the short summary and bullet points together with the final manuscript.

Referee #2:

The authors have addressed many of the original comments accurately, however, I still have two (minor) concerns:

1) Regarding map generating process: The added explanation in the method section is good, though it still lacks a description of the generation of the composite maps. At the moment, it reads as if the maps generated in the local refinement jobs themselves were used, rather than a composite map generated from two locally refined maps.

This is also true of the workflow graphics in Figure EV1 and Figure EV4; they both lack a mention of any composite maps. These should be shown here for clarity, as well as local resolution estimates of these maps. In addition: The sentence added on page 18 reads as if the authors used the locally refined maps for modelling, rather than the composite map as mentioned in the comment in the "Point by point" response file. Please make sure this is clear in the manuscript. The authors mentioned the composite maps on page 20, but still why they were made and for what purpose has not been mentioned yet. If these composite maps were used for modelling further detailed explanation of this process (what software was used) and reasons for it are required.

We thank the reviewer's comment. We have incorporated the composite map figures into the workflow in Figures EV1 and EV4. Additionally, we have included an explanation in the Methods section detailing how the composite map was generated and what local resolution estimates are.

In pages 17-18; For the hNPSF-NPFFR2-G_i-scFV16 complex, the local resolution of NPFFR2 is 3.4 Å, while the G-protein-scFV16 region reaches 3.0 Å. In the NPFFR2-BRIL-Fab^{BRIL}-Nb^{Fab} complex, the local resolution of NPFFR2 is 3.6 Å, and the BRIL-Fab^{BRIL}-Nb^{Fab} region achieves 2.8 Å. The locally refined maps were integrated using the 'vop maximum' function to generate each composite map, which was used for representation in Fig. 1B and Fig. 5A.

2) Regarding stubbing/removing of residues for deposition: It seems highly unusual to keep only some of the side chain atoms and not all/none. E.g. for the case of K268 and Q125 I have not seen this for any PDB files before, and the role of this residue and region seems rather over-interpreted, given the resolution. I would recommend to the authors to stub/remove the side chain and adjust the manuscript text accordingly. E.g. speculations/hypotheses for lower resolution regions are fine, however, making sure that the text clearly reflects the resolution limitations, and also making sure that if the remainder of the atoms were to be modelled, that these would make sense in terms of geometry/chemical restraints or clashes. If the side chains are to be stubbed, then the authors should make sure that the manuscript refers to additional data to support their hypothesis (e.g. functional readouts, MD simulations, etc).

As suggested by the reviewer, we have removed the side chains of Q125 in the ligand-free state and K268 in the active state. Accordingly, we modified Figure 5D and Appendix Figure S12 to show Q125 with its side chain removed in the ligand-free state, and revised the manuscript to delete the description of the Q125 rotamer change upon activation.

In page 11; In the ligand-free state, the density of this side chain is less resolved (Appendix Fig. S12). Along with Q125^{3.32}, V129^{3.36} is also involved in the interaction with F(-1)^{SF}, inducing an upward movement of TM3 by approximately 2 Å (Fig. 5D).

Regarding K268 side chain in the active state, we removed its side chain due to the lack of side chain density and updated Appendix Figure S2 accordingly. We also modified the manuscript to reflect this change.

In page 6; In the structure of NPFFR2 coupled to G α_i , F354^{H5.26} of G α_i interacts with NPFFR2 via van der Waals contacts with the C β carbon of K268^{6.29}; however its side chain is not fully resolved in our structure.

Furthermore, we have deposited the updated PDBs and now have updated validation reports. We have also revised the statistics table in Appendix Table S2.

Prof. Hee-Jung Choi
Seoul National University
Biological Sciences
1 Gwanak-ro, Gwanak-gu
Seoul, Seoul 08826
Korea, Republic of

Dear Prof. Choi,

I am very pleased to accept your manuscript for publication in the next available issue of EMBO reports. Thank you for your contribution to our journal.

Yours sincerely,
